# Deep learning at the edge enables real-time streaming ptychographic imaging

Anakha V. Babu[1,4,5], Tao Zhou[1,5], Saugat Kandel [1], Tekin Bicer [1], Zhengchun Liu[1], William Judge[2], Daniel J. Ching [1], Yi Jiang [1], Sinisa Veseli [1], Steven Henke [1], Ryan Chard[1], Yudong Yao[1], Ekaterina Sirazitdinova[3], Geetika Gupta[3], Martin V. Holt [1], Ian T. Foster [1], Antonino Miceli [1]✉ & Mathew J. Cherukara [1]✉

Coherent imaging techniques provide an unparalleled multi-scale view of materials across scientific and technological fields, from structural materials to quantum devices, from integrated circuits to biological cells. Driven by the construction of brighter sources and high-rate detectors, coherent imaging methods like ptychography are poised to revolutionize nanoscale materials characterization. However, these advancements are accompanied by significant increase in data and compute needs, which precludes real-time imaging, feedback and decision-making capabilities with conventional approaches. Here, we demonstrate a workflow that leverages artificial intelligence at the edge and high-performance computing to enable real-time inversion on X-ray ptychography data streamed directly from a detector at up to 2 kHz. The proposed AI-enabled workflow eliminates the oversampling constraints, allowing low-dose imaging using orders of magnitude less data than required by traditional methods.

Ptychography is a high-resolution coherent imaging technique that is widely used in X-ray, optical, and electron microscopy. In particular, X-ray ptychography has the unique potential for non-destructive nanoscale imaging of centimeter-sized objects[1,2] with little sample preparation, having provided unprecedented insight into countless material systems including integrated circuits[3] and biological specimens[4]. Strain information can be obtained additionally when combined with X-ray diffraction[5,6]. In the optical regime, the comprehensive depth information of ptychography has allowed 3D imaging of large and thick samples with micrometer resolution[7], while novel variations in the Fourier domain have enabled imaging in gigapixel scale[8] with single shot exposure[9]. Last but not least, recent developments in electron ptychography have witnessed the accomplishment of a record-breaking deep sub-angstrom resolution[10,11].

Ptychographic imaging is performed by scanning a coherent beam across the sample with a certain degree of spatial overlap and

recording the resulting far-field diffraction patterns. Subsequently, an image of the sample is recovered by computationally inverting these measured patterns. The inversion of ptychographic data (or phase retrieval) provides a solution to the phase problem, where only the amplitude information about the wave exiting the sample, and not its phase, is retained in the measured intensities. Currently, real-time imaging with ptychography is limited by the use of conventional phase retrieval methods. These methods do not produce live results until after acquiring a few tens if not hundreds of diffraction patterns, despite substantial advancement in the reconstruction algorithms[12]. Moreover, the spatial overlap required for the numerical convergence limits the sample volume that can be imaged in a given amount of time and can cause extra damage in dose-sensitive specimens.

State-of-the-art ptychography instruments bring about new and prohibitive computational challenges with their drastically increased

[1]Argonne National Laboratory, 9700 S Cass Ave, Lemont, IL, USA. [2]Department of Chemistry, University of Illinois, Chicago, IL, USA. [3]NVIDIA Corporation, Santa Clara, CA, USA. [4]Present address: KLA Corporation, Ann Arbor, MI, USA. [5]These authors contributed equally: Anakha V. Babu, Tao Zhou. ✉e-mail: amiceli@anl.gov; mcherukara@anl.gov

data rate. For example, one raster scan across an area of 1 mm × 1 mm at 100 nm step size yields 200 TB of raw data with a moderate size detector of one million pixels in 16-bit dynamical range. This volume of data can be acquired in <24 h at a fourth-generation synchrotron source, thus presenting a data rate of 16 Gbps and requiring ~PFLOPs of computation to perform phase retrieval[13]. To address this challenge, scientists have increasingly turned to deep learning methods for data analysis. An emerging strategy is to replace conventional analysis techniques with much faster surrogate deep learning models[14–17]. Deep convolutional neural networks have been widely explored for coherent imaging and have been shown to outperform conventional compute-intensive iterative algorithms used for phase retrieval in terms of speed, and increasingly also in reconstruction quality, especially under low-light or other sparse conditions[15,17–22]. However, there has not been a demonstration of real-time coherent imaging via deep learning to date, either at the edge or through on-demand resources at high-performance computing (HPC) facilities.

In this work, using X-ray ptychography as a representative technique, we demonstrate an artificial intelligence (AI)-enabled workflow that can keep up with the current maximum detector frame rate of 2 kHz to achieve coherent imaging in real-time. This workflow uses HPC resources for the online neural network training and a low-cost, palm-sized embedded GPU system (the edge device) at the beamline for real-time phase retrieval. We establish that this workflow is able to achieve accurate ptychographic imaging under extreme experimental conditions such as very low spatial overlap and very high framerate (up to 2 kHz). Additionally, we show that the accuracy of the workflow improves over the course of the experiment through continual learning. The proposed workflow provides a simple and scalable solution to the ever-growing data rate with state-of-the-art ptychography, and can be easily extended to work with other imaging techniques at light sources and advanced electron microscopes. This will potentially allow real-time experimental steering and real-time identification of

experimental errors, saving significant effort and cost across a wide range of experiments.

## Results

### Real-time streaming ptychography imaging workflow

The overall workflow for real-time streaming ptychographic imaging, shown in Fig. 1, consists of three concurrent components: (1) measurement, (2) online training, and (3) live AI inference. Diffraction patterns are captured on a detector downstream of the sample while scanning a focused X-ray beam in a spiral pattern. At the end of each scan, the resulting diffraction data are sent to the HPC resources where phase retrieval is performed using conventional iterative algorithm[23,24]. While iterative phase retrieval has also been used to produce real-time feedback for ptychographic experiments, this is typically too slow for data acquired with state-of-the-art detectors running at their maximum speed (>1 kHz), which is going to be a common practice at the fourth generation synchrotrons. Instead, the results of iterative phase retrieval are cropped and paired with the corresponding diffraction patterns to provide labeled data for incremental training of a neural network. Hereinafter, one set of training data refers to one diffraction pattern paired with a cropped image of the iteratively retrieved phase. The trained neural network model is periodically sent to the edge device for low-latency inference with improved quality. The diffraction patterns are also streamed concurrently to the edge device at the beamline over the local area network by using a codec-based structured data protocol[25]. Updated with the latest trained model, the edge device infers on individual diffraction patterns, and streams the results back to the beamline computer, providing the users with stitched sample images in real-time. The workflow is entirely automated, as documented in a video recording provided in the Supplementary Information. We note that while the workflow currently relies on a modified version of PtychoNN[15], it can be replaced with any

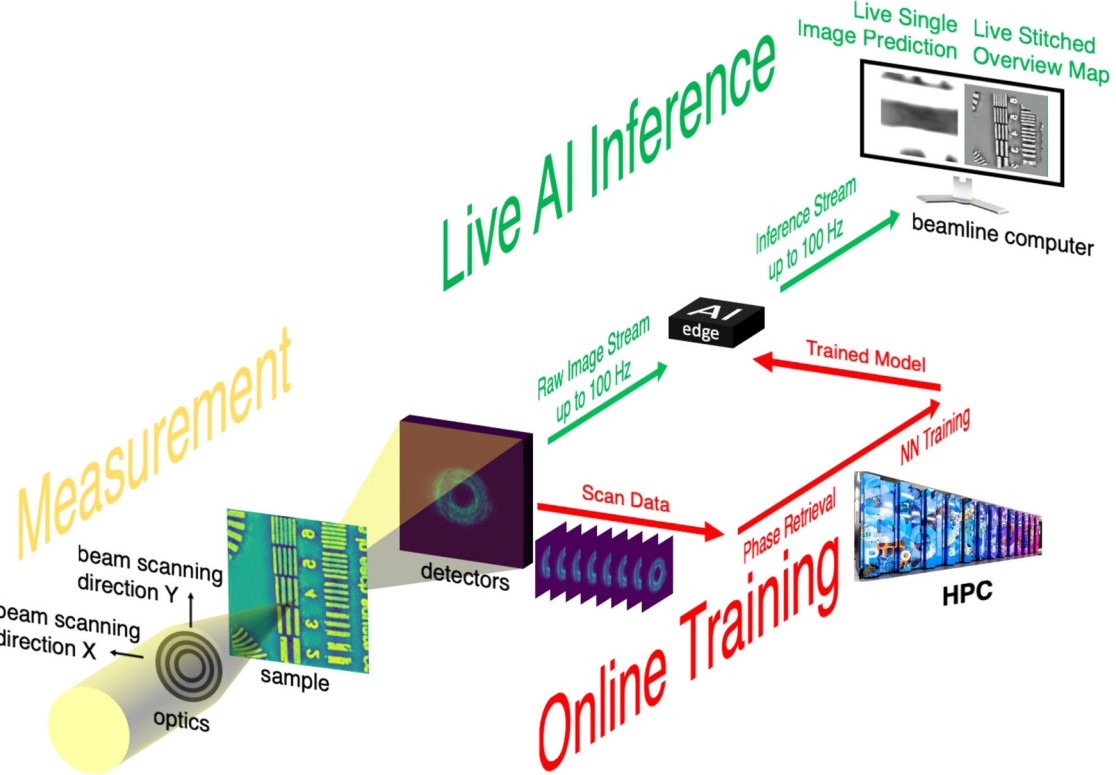

**Fig. 1 | Illustration of AI-enabled workflow for real-time streaming ptychography imaging.** An animated version of the sketch can be found here[27]. Image by Argonne National Laboratory.

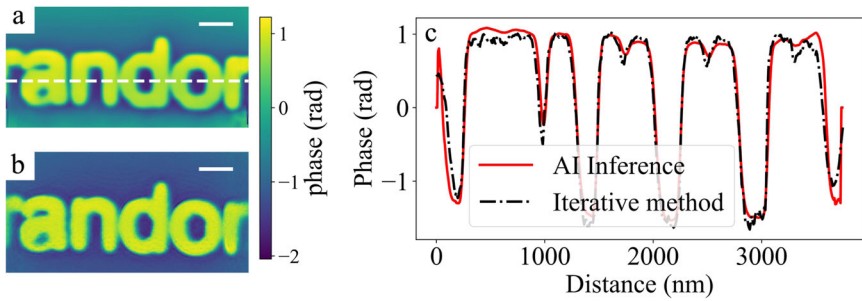

**Fig. 2 | Accuracy of AI inference compared to conventional iterative phase retrieval. a** Shows the cumulative phase from the workflow. The result is obtained by stitching together individual AI inferences of the entire scan as described in the "Methods" section. **b** Shows the results of iterative phase retrieval on the same data.

**c** Shows the line profile comparison of the phase obtained with AI inference and iterative methods. The position of the line cut is indicated by the white dashed line in (**a**). The scale bar is 500 nm. Source data are provided as a Source data file.

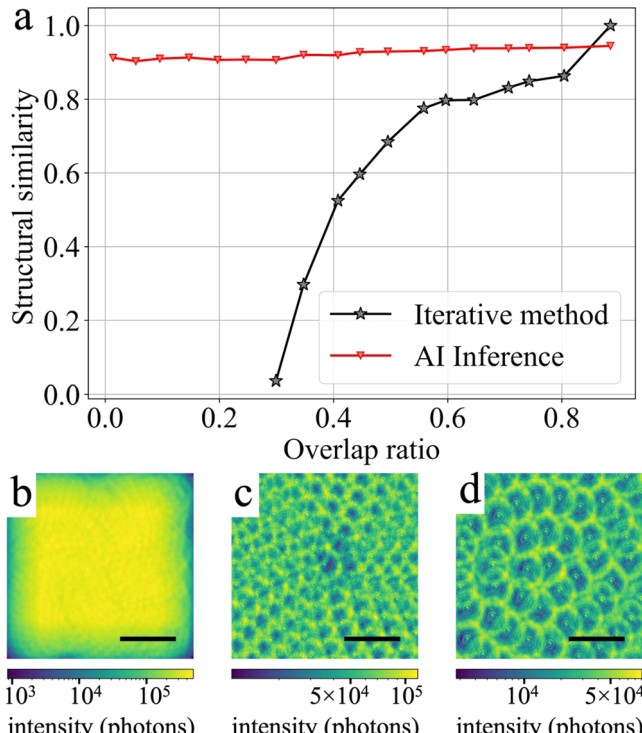

**Fig. 3 | Accuracy of AI inference on sparsely sampled data. a** Shows the accuracy of both AI inference and iterative phase retrieval, as a function of the overlap ratio. Also shown are visualizations of the actual probe overlap for overlap ratios of **b** 0.9, **c** 0.6, and **d** 0, by plotting the photon dose at each scan position. The average dose is 20k ph/nm² for an overlap ratio of 0.9, 1600 ph/nm² for an overlap ratio of 0.6 and 250 ph/nm² for an overlap ratio of 0. A defocused probe is used, resulting in a donut-shaped illumination on the sample that can be individually discerned at low overlap. The scale bar is 1 μm. Source data are provided as a Source data file.

alternative neural network in a plug-and-play fashion. More detailed information can be found in the "Methods" section.

**AI inference accuracy at the edge**
We first consider the ability of the workflow to reproduce results from iterative phase retrieval under the same experimental conditions. Figure 2 compares the AI-inferred and iteratively reconstructed phases for a test scan with high spatial overlap. A spiral scan was used with a step-size of 50 nm and a beamsize of 800 nm, corresponding to an overlap ratio of 0.9. Here the overlap ratio is defined as $1 - \sqrt{3}S/B$, where $S$ is the step-size of

the spiral scan and $B$ is the beamsize. Comparison of the line profiles in Fig. 2c shows that, once trained, AI inference results are almost identical to those obtained with iterative phase retrieval. Additionally, we note that the scanned area has features (alphabetical letters) that were not present in the training dataset. Supplementary Fig. 1 shows an example of the data included in the online training of the neural network which consists of entirely random patterns, although their refractive indices remain the same as the training and test data merely correspond to different regions of the same sample etched into different features.

**Low-dose ptychographic imaging through sparse-sampling**
We then explore the possibility for the workflow to invert sparsely sampled data. Because AI inference is performed independently on each diffraction pattern, the notion of spatial overlap no longer applies. A spiral scan with an overlap ratio of 0.9 is used as the starting point, the iteratively retrieved phase of which serves as the ground truth. We then gradually reduce the overlap by selectively removing part of the data. At each step, a new iterative phase retrieval is performed alongside the stitching of the AI inference results. The accuracy is evaluated as the structural similarity[26] of the iteratively retrieved or AI-inferred phase (see Supplementary Fig. 2) against the ground truth. As shown in Fig. 3a, while the stitched AI inference retains >90% accuracy even without any overlap ($S = 460$ nm, Fig. 3d), the accuracy of conventional iterative methods drops rapidly with decreasing overlap ratio, to <80% at an overlap ratio of 0.6 ($S = 180$ nm, Fig. 3c). It can thus be concluded that for an acceptable accuracy of 80%, the step size required for the workflow can be 2.5× the size required for conventional iterative methods. This in turn indicates that for a given amount of time, the workflow can cover 6.25× the area measured by conventional ptychographic imaging while simultaneously lowering the dose by the same factor on the sample. The latter is particularly appealing for dose-sensitive samples such as biological materials.

**Robustness in low light**
The beam dose can be further reduced by lowering the exposure time of the experimental data. Because the accuracy of the workflow is set by that of the trained model, the count rate of the training data is kept high. We demonstrate two strategies to address the disparity in the count rate between the training and the experimental data. The more straightforward solution is to scale up the experimental intensity before sending it to the edge device. This approach has the advantage of not requiring retraining of the neural network but only works with moderate scaling factors due to amplification of the noise-to-signal ratio. Figure 4a shows an example of the experimental data with an

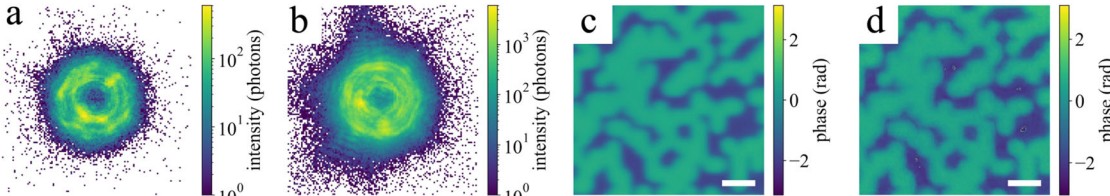

**Fig. 4 | Accuracy of AI inference on low count data. a** Shows an example diffraction pattern of a ptychographic scan with an exposure time of 0.5 ms. **b** Shows an example diffraction pattern of the training data with an exposure time of 5 ms. **c** Shows the stitched AI inferred phase on the low count dataset shown in (**a**), obtained after upscaling the experimental intensity by a factor of 10. **d** Shows the iteratively retrieved phase of the low photon count dataset shown in (**a**). The scale bar is 500 nm.

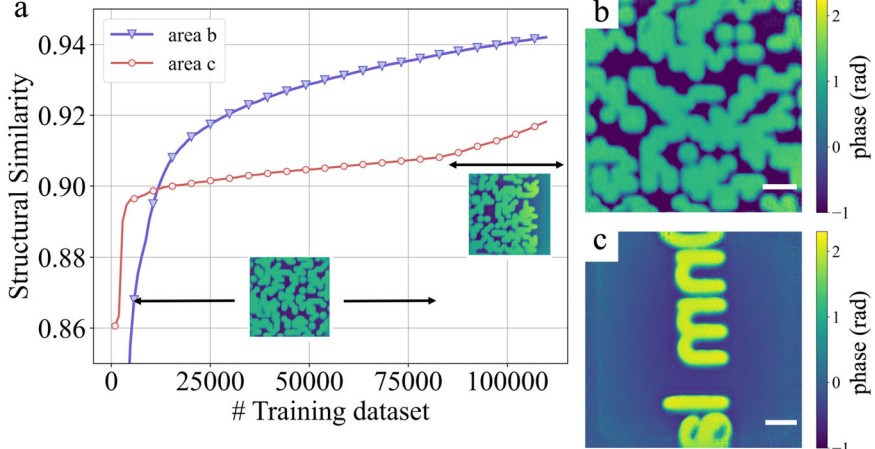

**Fig. 5 | Effect of continual learning. a** Evolution of structural similarity against the ground truth for two selected test-sample areas during continual learning. Those areas are shown respectively in (**b**) and (**c**). The training set consists of pairs of iteratively retrieved phases and corresponding diffraction patterns on an area with randomly etched features. After about 80,000 sets, the training data starts to include the edge of the patterned areas. The scale bar is 500 nm. Source data are provided as a Source data file.

exposure time of 0.5 ms, corresponding to the maximum detector frame rate of 2 kHz. Figure 4b shows an example of the training data, exposed for 5 ms. Figure 4c shows the stitched AI inference results after upscaling the experimental intensity by a factor of 10, with an accuracy of 86% as measured against the groundtruth image reconstructed by iterative phase retrieval (Fig. 4d). A video recording of the live demonstration of inference at 2 kHz can be found here[27]. The estimated dose for this demonstration was 6 ph/nm². The second solution is to scale down the intensity in the training data and retrain the model. This way a much larger scaling factor can be achieved. No noticeable difference was observed in the AI inference whether or not Poisson noise was added after scaling down the intensity. As is shown in Supplementary Fig. 3, an accuracy of over 80% is observed with a scaling factor of as large as 10,000.

### Effect of continual learning

To ensure a high accuracy of the AI-inferred phase, and to quickly adapt to new sample features, the neural network model at the edge is constantly updated through continual learning. Figure 5a shows the performance of the workflow on two test-sample areas, shown respectively, in Fig. 5b and c. The first area contains features similar to the training data, in which case the accuracy of the training improves rapidly, and highly accurate AI inference is achieved using just the first 20,000 sets of training data. The second area contains features unseen in the training data. The accuracy in this case improves progressively over the course of the continual learning. The change of slope in the structural similarity after 80,000 sets is explained by added diversity, as the training data at this point starts to include edges of the patterned areas. The continual learning strategy is thus essential to achieve accurate AI inference on new features. The reconstruction fidelity of the neural network at the initial stage of continual learning is illustrated in Supplementary Fig. 4.

### Discussion

The workflow described in this work provides a path to tackling the data and compute needs for ptychographic imaging at next-generation light sources and at advanced electron microscopes. As shown in Fig. 6, the raw data acquisition rate for both state-of-the-art X-ray[28–30] and electron[11] ptychography doubles every year, enabled by the advancements in both scanning strategies and larger and faster detectors[31–35]. For X-ray ptychography, the raw data rate reached 800 Mbps a decade ago, and this still exceeds the throughput of the most advanced iterative optimizers published in 2021[36,37]. To put this into perspective, it would take conventional methods an hour to perform phase retrieval on data taken in a second on a fully illuminated 10 Mega-pixel detector running at 32 bits and 2 kHz (640 Gbps). Our workflow overcomes this by leveraging the low-latency, high-throughput surrogate models at the edge. The inference speed in our demonstration is ultimately limited by the 1 Gbps network connection on the detector control computer. For a detector image size of 512 × 512 pixels, live inference at 100 Hz is achieved, corresponding to a capped incoming data rate of 0.5 Gbps. By reducing the image size to 128 × 128 pixels, live AI inference at 2 kHz is achieved while running the detector at its maximum frame rate. Using a powerful GPU, the inference time was further reduced to 70 µs per image (see the "Methods" section), corresponding to a frame-rate of 14 kHz. For comparison, 500

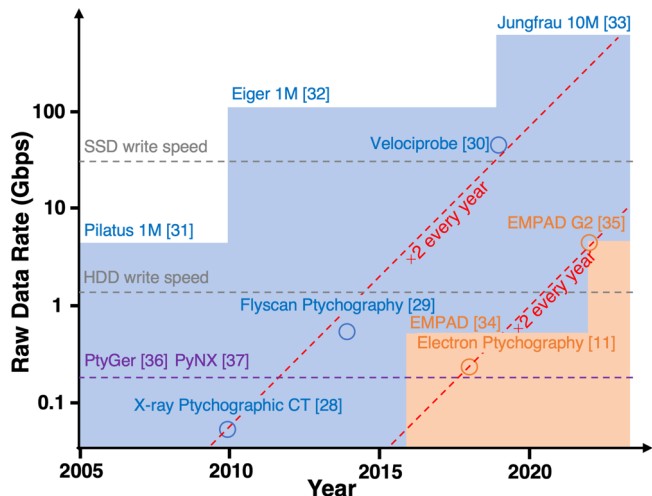

**Fig. 6 | Evolution of raw data rate for state-of-the-art X-ray and electron ptychography.** Source data are provided as a Source data file.

iterations of iterative phase retrieval took about 1 min for a scan consisting of 963 images, corresponding to an equivalent frame-rate about 1000 times slower, at 16 Hz.

Given sufficient data, the accuracy of the workflow improves over the course of the experiment via continual learning and is ultimately limited by the accuracy of the iterative phase retrieval used for the training. Because access to on-demand HPC resources may be limited, a strategy is devised to minimize the use of expensive HPC resources. In the initial phase, when a new set of experimental data is received at the HPC, it automatically triggers the image reconstruction service using iterative methods. The results are then cropped and paired with the corresponding diffraction patterns to form the labeled training data. Part of this training data is used immediately to validate the accuracy of the existing neural network. If a large mismatch is detected between the AI-inferred phase from the latest model and the iteratively retrieved phase in the training data, a retraining service is queued which upon completion also sends the updated model to the edge device. If the mismatch is within a certain tolerance level (for instance, a difference of <10% in structural similarity), the automatic image reconstruction and retraining services are suspended, with the experimental data being sent and reconstructed at HPC periodically at a prolonged interval (for instance, once every hour). Comparing the AI inference using the existing neural network with the periodically retrieved phase is particularly useful to check, among others, data distributional shifts. The on-demand image reconstruction and retraining services are resumed if a large mismatch is again detected. Typically, a new network is trained from scratch at the beginning of each experiment to better accommodate for differences in samples, probe, and detector configuration However, more sophisticated schemes such as the one described in the fairDMS approach[38] can be adopted to reuse models from previous experiments for achieving more rapid training of the data.

One of the reasons behind the high inference accuracy demonstrated in this work is that the neural network is trained specifically for samples with a given range of refractive indices and for a fixed illumination probe. This simplifies significantly the intensity (-input) phase (-output) relationship for the neural network, thus allowing it to accurately solve the inverse problem. As long as the refractive index of the sample remains in the same range as those used in the training data, the neural network can accurately predict new features it has never seen such as those demonstrated in Fig. 2. If the refractive index of the sample falls outside of the range however, a retraining of the neural network is recommended. We note that it is not necessary to retrain neural networks for small fluctuations in the sample or in the

probe. In fact, the neural network has been shown to be quite tolerant to those changes. In Supplementary Fig. 5, we use the same model to directly infer on experimentally acquired data with different count rates from the same area. The stitched inference for when the count rate has varied by a factor of 4 shows no visible difference compared to the expected result. When the count rate has varied by a factor of 16, the predicted sample structure still looks correct, but the predicted level of phase is off by as much as ±0.2 rad. When the count rate has varied by a factor of 40, even the predicted structure is wrong, and the stitched inference is no longer usable. This shows that the trained model can still produce reliable phase inference even if the count rate has varied by as much as a factor of 4. The predicted phase will still be usable as feedback for experimental steering[39] even if the incoming intensity has varied by as much as a factor of 16.

One significant advantage of the proposed workflow is in the field of low-dose high-resolution imaging, which is relevant for a variety of materials including biological samples and organic–inorganic perovskites. Conventional ptychography is in particular damaging to these materials due to the beam exposure with overlapping constraint. Beam damage is an even bigger problem for electron ptychography than for X-ray due to the stronger interaction with materials of the electron beam. In this work, low-dose high-resolution imaging is achieved through two separate approaches. First, by performing AI inference on individual diffraction patterns, the oversampling constraint is eliminated. We show a sample image with an accuracy of 90% without any probe overlap between the measured points. This reduces the beam dose by a factor of 6 compared to that required by conventional methods. In the second approach, the beam dose is further reduced by a factor of 10 (upscaling the low-count rate experimental data, Fig. 4) to 10,000 (downscaling the training data taken at higher count rates, Supplementary Fig. 3) by performing AI inference on data acquired with shorter exposure time. The upscaling method has the advantage of working with low-count data, while the downscaling method is convenient as it does not require retraining of the existing model. Both approaches, in addition to reducing beam damage by a few orders of magnitude, increase the measurement area per given time by the same amount. We note that the proposed workflow still requires high-overlapped data as well as iterative phase retrieval to produce training data for the neural network, particularly at the beginning of the experiment. As such, a good strategy for imaging beam-sensitive samples is to first run conventional ptychographic measurements in a small area with high overlap and a decent count rate. This allows high-quality training data to be acquired, which ensures the accuracy of the subsequent low-dose imaging on the remaining area with essentially no overlap. The limitation of this strategy is that by removing the overlap between the measured points, we have limited the possibility to perform iterative phase retrieval, and by extension the possibility to perform continual learning.

## Methods
### X-ray ptychography experiment
X-ray ptychography data was taken at the hard X-ray nanoprobe beamline at the Advanced Photon Source. The photon energy was 10 keV. Two sets of experimental data were acquired. The first set was taken with an Amsterdam Scientific Instruments Medipix3 detector (516 × 516 pixels, 55 µm pixel size) sitting at 1.55 m downstream of the sample. The maximum frame rate was 100 Hz limited by the data transfer bandwidth of the 1 Gbps network. The second set was taken with a Dectris Eiger2 X 500K detector with 75 µm pixels, located 0.9 m downstream from the sample. Only a 128 × 128 subsection of the image is used, allowing live inference at the maximum detector frame rate of 2 kHz. A Fresnel Zone Plate with 160 µm diameter and 30 nm outermost zone width was used and defocused intentionally to reach a spot size of about 800 nm. For each scan a piezo motor moves the focusing optics, and hence the beam, in

963 steps following a spiral pattern. The step size varies from 50 nm (for training) to 200 nm while the exposure time varies from 0.4 s (for training) to 0.5 ms.

## Iterative phase retrieval and image processing toolkit

Iterative phase retrieval was performed using the regularized Ptychographic Iterative Engine (rPIE)[23] algorithm implemented in the Tike toolkit[40] (version 0.22.2). Tike provides an application programming interface (API) to implement, parallelize, and run different types of 2D ptychographic reconstruction algorithms, including PIE family and the difference map (DM) algorithms. We have also performed phase retrieval using DM to compare with those retrieved with rPIE shown in the manuscript. As shown in Supplementary Fig. 6. The phases retrieved with both methods are mostly consistent with each other. The phase retrieved with DM is slightly less uniform, which is why we have kept the rPIE retrieved phase for this work. All the reconstructions shown in Supplementary Fig. 6 were performed on an RTX 2080 Ti GPU, with 3 probe modes and 2000 iterations.

Tike supports high-performance scalable reconstruction operations via accelerators (GPUs) and optimized communication primitives tailored to ptychography[36,41]. The optimizations address the performance bottlenecks with the process and/or thread synchronizations during the overlapping (or halo) region updates using better communication and task placement topologies. These optimizations enable efficient multi-GPU and multi-node data processing that can scale on high-end compute clusters and supercomputers.

During our experimental evaluation, we used ThetaGPU cluster at Argonne Leadership Computing Facility (ALCF). ThetaGPU cluster consists of 26 NVIDIA DGX nodes, each of which is equipped with eight A100 GPUs (connected with an NVLink switch), 360 GB memory, and an AMD EPYC 7742 processor. We reserved eight GPU nodes, i.e., a total of 64 A100 GPUs, to perform phase retrieval and model training. The updated trained models are periodically sent to the edge device at nanoprobe beamline at APS. ALCF and APS facilities are connected with a 200 Gbps link. We initiated model training and phase retrieval operations periodically during data acquisition. We used 7 DGX nodes (52 GPUs) for concurrent phase retrieval operations and the remaining node (8 GPUs) for PtychoNN model training. We coordinated the workflow components with Globus Automate/Flows (for workflow definition), funcX[42] (for remote function calls and resource management), and Globus (for inter-facility data movement)[43,44].

## PtychoNN 2.0 architecture

The workflow uses a convolutional neural network that takes as input a raw coherent diffraction image and outputs an inferred sample structure in a single pass (live inference). In this paper, we use a modified version of PtychoNN[15] (PtychoNN 2.0) for live AI inference to achieve low inference latency at the edge. The detailed architecture of PtychoNN 2.0 is shown in Supplementary Fig. 7 and this network differs from PtychoNN in two different aspects. First, it predicts only the phases and therefore does not contain the amplitude branch originally present in PtychoNN. Second, each convolutional layer in PtychoNN 2.0 contains 0.5X the number of filters as PtychoNN. With these modifications, the number of trainable parameters in PtychoNN 2.0 is reduced to 0.7M parameters against 4.7M parameters in PtychoNN. Reducing the model size has the obvious advantage of improving the training times and also helps in lowering the inference latency at the edge without significantly impacting the accuracy. The performance of the two ML models is analyzed in terms of the quality of the phase predictions and inference times. Supplementary Fig. 8 indicates that PtychoNN 2.0 can give equally good predictions as PtychoNN.

As an edge device, NVIDIA's AGX Xavier supports native PyTorch implementations, we compared the inference times for PyTorch against TensorRT for both ML models. Table 1 shows the average execution times observed on AGX, over 50 iterations for

**Table 1 | Approximate inference times (ms) in TensorRT and PyTorch on the AGX**

| Model # | TensorRT | PyTorch |
|---|---|---|
| PtychoNN | 10 ± 1 | 15 ± 1 |
| PtychoNN 2.0 | 2.3 ± 0.4 | 8 ± 1 |

each of the models in TensorRT and PyTorch for a batch size of 1. The AGX was operated in Max-N mode during these inference runs. It has to be noted that the inference times shown in the table exclude the pre-processing time. PtychoNN in Table 1 corresponds to the network discussed in ref. 15 and PtychoNN 2.0 refers to the lightweight model designed for faster ptychographic reconstructions. TensorRT is observed to exhibit a speed up of approximately 4× when compared to native PyTorch implementation for PtychoNN 2.0.

## Neural network training

Compared to other works employing neural networks for phase retrieval[45–48], training is performed notably online in this workflow. In other words, the neural network is constantly being updated during the experiment as new diffraction data are acquired. Iterative phase retrieval is performed on the HPC each time a new batch of data is available. The diffraction data plus the phase-retrieved images are appended to the existing corpus of training data, and the neural network is trained for a further 50 epochs by using a cyclic learning rate policy[49]. Network weights are updated by using the adaptive moment estimation (ADAM) optimizer to minimize the mean absolute error (MAE) between the target labels (output of iterative phase retrieval) and AI inferences. The model state at each training epoch is evaluated on unseen validation data, which is 10% of the training data. The validation data set is randomly chosen from the training data for every epoch. The trained model with the lowest validation loss at the end of 50 epochs is pushed to the edge device for the AI inference on subsequent scans. During the inference, the neural network will infer from unseen test images that are not present in training or the validation datasets. Supplementary Fig. 9 shows the training and validation losses for the first training iteration with only one ptychographic dataset (scan #1) and for training iteration 111 with 111 ptychographic datasets (when scan #111 was added). While we do not see evidence of overfitting in the early iterations, we notice that the training loss slightly diverges in the latter iterations. Here, the cyclic learning rate approach was used at each training iteration to ensure that we are not stuck in a local minimum even as we add more training data. A total of 113 experimental scans, each with 963 X-ray diffraction patterns were acquired. At the end of the experiment, the total available training data comprised 113 × 963 = 108,819 pairs of diffraction and sample images. As is shown in Supplementary Fig. 10, the training time scales almost linearly with the size of the training data. For 100,000 pairs of data, the training time is about 10 min using 8 A100 GPUs at the ALCF.

Each diffraction pattern represents a certain field-of-view of the sample illuminated by the probe at that scan position. We select this field-of-view based on the FWHM of the probe. For a reconstructed pixel size of 6.82 nm and a probe FWHM of 800 nm, an area of 128 × 128 pixels was cropped each time to be paired with the corresponding diffraction pattern for the training. Because the measurements were performed with a spiral scan on an irregular scan grid, instead of grabbing data directly from the iteratively retrieved result, the phase in the training data was interpolated on a regular grid with 128 × 128 pixels centered around the center of the beam illumination.

## Stitching of AI inference results

The workflow produces, for each input diffraction pattern, an inference image of 128 × 128 pixels. Each inference image is centered around the corresponding beam position on the sample which, for

spiral scans, is not on a regular grid. The first step of stitching is thus to interpolate all the AI inference results onto the same regular grid, based on information such as the beam position and the pixel size of the inference image. The stitched image is calculated by statistically averaging all the interpolated results on the same but larger regular grid. Alternatively, we have performed a weighted average of the interpolated results using the probe amplitude as the weights. The second approach yielded very comparable results as the original one with no visible improvement in the quality of the final image.

### Estimation of the spatial resolution

In order to estimate the spatial resolution of the iterative retrieved phase and the AI-inferred phase, we have picked two scans that share a large overlap area. We note that despite covering the same area, the beam positions on the sample for these two scans were completely different. Supplementary Fig. 11a and b shows the iteratively reconstructed phase of the two scans, with the overlap area highlighted by the red rectangle. We have performed Fourier shell correlation calculation on the overlapped area, the resolution of iterative phase retrieval was about 35 nm, as indicated in Supplementary Fig. 11c. The beam size on the sample was about 800 nm for those measurements. Next, we perform NN inference on these two scans. For this test, we retrained the NN after specifically removing those two scans from the training data. The stitched inferences for the two scans are shown in Supplementary Fig. 11d and e, with the overlap area highlighted in red. Fourier shell correlation (Supplementary Fig. 11f) performed on the overlap area of the two stitched inference images indicates a resolution of 20 nm, much better than the number obtained for iterative phase retrieval. We are not confident this reflects the actual resolution of the NN inference and have hence chosen not to include this in the main manuscript.

### NVIDIA's Jetson AGX Xavier developer kit

We used the NVIDIA Jetson AGX Xavier developer kit[50], an embedded GPU platform, to demonstrate real-time ptychographic phase retrieval at the edge. AGX Xavier is one among the Jetson series from NVIDIA with a computational capability of 32 TOPs, dedicated to building embedded ML solutions. Here the AGX Xavier is targeted as an inference device and the TensorRT$^{TM}$ Python API from NVIDIA is used for accelerating the inference workflow. One of the primary means of importing the model in TensorRT$^{TM}$ is via the ONNX format, and therefore the PyTorch-trained model is converted to ONNX before running the inference workflow.

### Real-time phase retrieval at higher frame rates

The embedded GPU device (NVIDIA Jetson) can support real-time feedback up to a frame rate of 100 Hz. Data acquisition rates for ptychography experiments are growing rapidly and are soon expected to exceed 50 Gbps. Therefore, it is important to reduce the AI inference times to achieve real-time data analysis. Widely studied and adopted techniques like reduced precision arithmetic and quantization can be used to accelerate AI inference[51]. However, such methods can degrade network accuracy and require careful fine-tuning of hyperparameters. Since the neural network in the study, PtychoNN 2.0, is a fully convolutional neural network and is compute-limited, we explored the use of an advanced GPU that is designed to accelerate ML workloads. We benchmarked the AI inference time of an Ampere-architecture GPU (RTX A6000) for high-speed inference with different batch sizes as shown in Table 2. The Dectris Eiger2 X 500K detector used for high-speed data acquisition can support a maximum data rate of 2 kHz at 16 bits. A batch size of 8 was chosen for the AI inference at 2 kHz as the inference time per frame is observed not to improve substantially for higher batch sizes. Table 2 shows the average AI inference time, plus standard deviations, as measured across 50 inference runs.

**Table 2 | Approximate inference times (µs) per frame on the RTX A6000 & AGX Xavier**

| Batch size # | RTX A6000 | AGX Xavier |
|---|---|---|
| 1 | 370 ± 20 | 2300 ± 400 |
| 2 | 220 ± 20 | 1360 ± 340 |
| 4 | 130 ± 10 | 960 ± 110 |
| 8 | 90 ± 5 | 850 ± 140 |
| 16 | 70 ± 5 | 680 ± 30 |

## Data availability

The ptychographic experimental data used in the inference is publicly available at https://doi.org/10.5281/zenodo.8121606. The source data underlying the plots contained in this article is also provided as a Source Data file Source data are provided with this paper.

## Code availability

The code and machine learning model for the inference implemented at the edge is available at https://github.com/vbanakha/edgePtychoNN.

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

## Acknowledgements

This research used resources of the Advanced Photon Source (APS), Center for Nanoscale Materials (CNM), and Argonne Leadership Computing Facility (ALCF), which are operated for the DOE Office of Science by Argonne National Laboratory under Contract No. DE-AC02-06CH11357. This work was also supported by the U.S. Department of Energy, Office of Science, Office of Basic Energy Sciences Data, Artificial Intelligence, and Machine Learning at DOE Scientific User Facilities program under Award Number 34532. M.J.C. and S.K. also acknowledge support from Argonne LDRD 2021-0090—AutoPtycho: Autonomous, Sparse-sampled Ptychographic Imaging. We gratefully acknowledge the computing resources provided on Swing, a high-performance computing cluster operated by the Laboratory Computing Resource Center at Argonne National Laboratory. We gratefully acknowledge insightful discussion and advice from Francesco De Carlo at the Advanced Photon Source. We would also like to thank William Allcock and his team at the ALCF for their help with the HPC resources used in this work, including the on-demand computing testbed.

## Author contributions

A.V.B., T.Z., S.K., A.M., and M.J.C. conceived the workflow. A.V.B., T.Z., and S.K. developed the workflow software with the assistance of T.B., Z.L., W.J., Y.J., S.V., Y.Y., and I.T.F.; T.Z., A.V.B., S.K., A.M., and M.J.C. performed the synchrotron experiment with the assistance from M.V.H.; A.V.B., T.Z., S.K. performed the data analysis under the guidance of A.M. and M.J.C.; E.S. and G.G. provided assistance on the use of the edge computing hardware. D.J.C., S.H., and R.C. provided assistance on the

use of high-performance computing hardware. All the authors discussed the results and wrote the manuscript.

## Competing interests

The authors declare no competing interests.
