## [Peer Review File · Nature Communications]

Deep learning at the edge enables real-time streaming
ptychographic imagingReviewer #1 (Remarks to the Author):

This paper describes a workflow combining high performance computing (HPC), deep learning, real-time inference on edge devices, to improve ptychographic imaging at a beamline, resulting in improved phase retrieval at lower doses and overlaps than using conventional algorithms.

This is an important paper in many respects. It is one of the few papers that connects beamline microscopy to HPC systems in an automated manner that includes a loop from experimental results. The former (connecting HPC to microscopes) is not necessarily new, but the latter most certainly is. The authors achieve this via continual training of a neural network, and continual evaluation of the performance of the neural net compared to the training data, to determine whether a new round of training is necessary for satisfactory performance. As such, this paper's novelty to me is that it shows the benefits of an automated, computational workflow in concert with a controllable experiment, yielding tangibly better results than the current state of the art.

I have just a few questions / comments that I hope the authors will be able to address:

1. To what extent are the trained models reusable? Can a model be trained on a single sample, then used on the same sample after a few weeks (given drift in parameters)?
2. Is there any uncertainty quantification (even using something simple like dropout) that could be implemented for the algorithm? One of the concerns here is that if there is an underlying distributional shift, then lack of UQ can mask errors in the inference, so have some UQ would be advantageous.
3. On the question of distributional shift, is this something that could occur during the experiment itself? How would one detect it if so and deal with it? (I can think of examples, such as ensemble learning or cycle-GANs that could be used to counter it).
4. Is there an option to incorporate simulations into this workflow in the future?

Reviewer #2 (Remarks to the Author):

In this article, Babu et al. apply a deep convolutional neural network (i.e., PtychoNN 2.0, modified from PtychoNN) for real-time ptychographic imaging. The presented Machine Learning (ML) approach demonstrated the possibility of the real-time ptychographic image. However, while the authors claim in the article that the iterative method is slow, the proposed ML approach still needs the iterative reconstruction method. This will limit the application of ML-based approach and also affect the general of the proposed ML approach.

Moreover, despite the interest in applying a modified deep neural network for real-time application, the discussion about the comparison between the iterative phase reconstruction and the ML-based method shows that the proposed ML approach gives a worse reconstruction compared to the iterative method. As shown in Fig. 2C, Fig. 3E, and Fig. 3F, the ML approach cannot recover the fine feature of the measured sample. Even there is a speed advantage for the ML-based approach, reconstruction accuracy should be more important than reconstruction speed. Therefore, I would not recommend this work for publication in Nature Communication but in a more specialized journal focused on AI techniques, unless they can make a significant improvement in the performance of the ML-based approach over the iterative method.

Below are some additional questions:

1. The detailed architecture of the ML model is missing in the article, please clarify it in the paper.
2. What is the main difference between PtychoNN 2.0 and PtychoNN?
3. What is the reconstruction resolution for Figure 2, Figure 3, and Figure 4? The structural similarity can be used only when the ground truth is known. For ptychography, the ground truth is generally not known. As a widely adopted approach, the authors should use the Fourier shell ring method instead of structural similarity.
4. When the different counts of diffraction data from the same region are given to the model, will the ML model give the same results or different results?

5. How is the stitching of individual reconstructed images from the ML model performed? Please give the details.
6. The image size for all the Figures in the paper are missing.
7. When the focus of the used X-ray is changed or off, will the ML model give the same results?
8. Did the author try the Difference Map (DM) method for these experimental data? To make a fair comparison, the author should also use the DM iterative method.
9. When the training data size is increasing, will the model need a longer time to finish online training?
10. What method is used to avoid the overfitting of the ML model? Why did you choose 90% of the data as training and 10% as validation? Did you have separate images for testing or your validation and test are the same?

Reviewer #3 (Remarks to the Author):

The authors present a neural-network framework for live ptychographic reconstruction feedback. The proposed framework claims that it can eliminate the overlap constraint in ptychography and enable low-dose imaging. The claims are supported by experimental results where the data was acquired at an impressive rate of up to 2kHz. The results show that the trained neural network reconstruction accuracy is as good as that of iterative phase retrieval, once all training data is taken into account. The neural network can also be pre-trained on different sample types (that meet specific criteria) which can be useful for practical experiments. There are also results showing the performance of the neural network for low-overlap scanning and low-exposure imaging. I believe this work has great potential for the 4th generation synchrotron beamlines where the radiation dose and the resulting sample damage will be ultimately limiting the resolution at which the samples can be imaged. Especially for ptychographic X-ray tomography. The prospect of low-overlap and low-exposure ptychography sounds very promising for low-dose, high-speed ptychographic imaging. The neural network is also a great candidate for high-speed data-processing which will only increase in the next generation X-ray sources.

However, there are several general remarks that still remain upon reading the manuscript:

1. While the introduced workflow is claimed to be very fast, I lacked information on how long it takes to train the neural network and how long it takes to reconstruct the data using a neural network. I would also like to see a comparison between the neural network training speed and iterative phase retrieval.

2. The authors could also be more transparent on the capabilities of the neural network. My impression was that the two imaged samples had the same refractive index. As I read the paper I discovered that there are significant constraints on the neural network pre-training such as the need to use samples with the same refractive index, the need for the same exposure times etc. Having a discussion on the practical capabilities of the pre-trained neural network would be useful to the reader.

3. The authors could also improve the presented results and claims of low-dose imaging using the neural network. For example, the neural network was trained on highly overlapping data, which was then used to reconstruct low-overlap data of the same sample field-of-view. Does the neural network have to be pre-trained on highly overlapping data? Is the accuracy of the pre-trained model maintained when imaging a different sample field-of-view with a reduced overlap? For example, to image a 1x1mm² area, can you image a micron sized area with high overlap and then image the remaining area with a virtually zero overlap? Addressing these comments would either improve the impact of the paper and/or inform the reader on the use-case of the proposed workflow.

4. There were also claims of low-dose imaging, but the authors never really stated the actual dose imparted onto the sample. While there are two sections describing low-exposure imaging and low-overlap imaging, in each of the cases the total dose incident onto the sample can be misleading to the reader. For example, low-exposure imaging with an overlap ratio of 99% can have the same dose as an experiment with a moderate exposure time and a moderate overlap. Since dose is

linked to both the exposure time and the overlap percentage more details are necessary to assess the importance of the proposed results for low-dose experiments.

Below you can find specific comments to parts of the manuscript:

1. In the Introduction you say: "The inversion of ptychographic data (or phase retrieval) provides a solution to the phase problem, where only the amplitude information about the sample, and not its phase, is retained in the measured intensities."

This sentence seems to suggest that the intensity of the measured diffraction patterns encodes the absorption of the sample, while the phase of the diffraction patterns encodes the phase of the sample. This is not true since the measured intensities (diffraction patterns) contain information of both the absorption and phase of the sample. Otherwise, phase only samples would have zero diffraction signal.

2. In the Introduction you say: "Currently, real-time imaging with ptychography is limited by the use of conventional phase retrieval methods. These methods do not produce live results until after acquiring a few tens if not hundreds of diffraction patterns, despite substantial advancement in the reconstruction algorithms."

Could you clarify what you mean by real-time imaging? There are X-ray synchrotron beamlines that can reconstruct ptychographic data faster than the data acquisition rate (acquisition taking up to a minute for high-resolution measurements). I would argue that current existing algorithms are fast enough for ptychographic tomography, so it depends what you mean by real-time imaging. Are you referring specifically to focused-beam ptychography? For example, near-field ptychography uses a huge probe and can capture only 10-20 diffraction patterns to cover a wide sample area where the reconstruction speed is limited not by iterative phase retrieval but rather by the I/O operations of the data.

Also the number of diffraction patterns is proportional to the sample field-of-view, so I don't really see the link with reconstruction algorithm advancements. If you want to reconstruct a meaningful field-of-view, the 10-100 diffraction patterns you are referring to must be captured for visualization to happen.

I think you should emphasize more the super high speed imaging at 2kHz that you are doing where the conventional phasing algorithms are definitely too slow.

3. In the "Real-time Streaming Ptychography Imaging Workflow" section the authors say: "Iterative phase retrieval is typically too slow for low-latency feedback even if the processing is performed using HPC resources."

I will disagree since this low-latency feedback is done routinely in beamlines currently without deep-learning phase retrieval methods. Define your criteria: 4th generation sources, low exposure imaging, 2kHz data collectio etc.

4. In the "Real-time Streaming Ptychography Imaging Workflow" section the authors say: "Instead, the results of iterative phase retrieval are cropped and paired with the corresponding diffraction patterns to provide labeled data for incremental training of a neural network." and "Hereinafter, one set of training data refers to one diffraction pattern paired with a cropped image of the iteratively retrieved phase"

Typically the diffraction pattern at a given scan point represents a certain field-of-view of the sample. By cropping, do you mean that you select a field-of-view area of the sample defined by the probe size and obtain a pair of field-of-view plus a diffraction pattern? Also when do you retrieve the recovered phase? After 1 iteration or more? Moreover, do you send the paired data for training after every single scan? More details would be helpful.

5. In the "AI Inference Accuracy at the Edge" section the authors say: "Comparison of the line profiles in Figure 2C shows that, once trained, AI inference results are almost identical to those

obtained with iterative phase retrieval.”

Image quality depends on iteration number. Could you also state how fast the AI reconstruction is compared to the iterative engine in this case? Quality vs speed is an important metric.

6. In the “AI Inference Accuracy at the Edge” section the authors say: “Additionally, we note that the scanned area has features (alphabetical letters) that were not present in the training dataset.

This is the first mention that the training does not have to be done for each individual sample. Instead, the network can be pre-trained and re-used on other sample types. It would be useful to make this clear in the introduction of the technique.

7. In the “AI Inference Accuracy at the Edge” section the authors say: “Supplementary figure S2 shows an example of the data included in the online training of the neural network which consists of entirely random patterns, although their refractive indices remain the same as the training and test data merely correspond to different regions of the same sample etched into different features.”

As I keep reading the text, I keep discovering where the neural network would and would not work. It would be useful for the reader to have a summary of what the neural network can do. For example, the pre-trained neural network can be used only on samples with the same-refractive index (as you state in this paragraph), the data must have the same exposure time (stated in the following paragraphs) etc.

8. In the “Low-dose Ptychographic Imaging through Sparse-sampling” section the authors say: “A spiral scan with an overlap ratio of 0.88 is used as the starting point, the iteratively retrieved phase of which serves as the ground truth. We then gradually reduce the overlap by selectively removing part of the data. At each step, a new iterative phase retrieval is performed alongside stitching of the AI inference results.”

As the overlap is decreased, you keep training the same network on the same sample. Since the neural network already knows how to do the phase retrieval of the highly overlapping data it seems to me that the phase retrieval task of the data with low overlap becomes much easier, hence the high-accuracy results. For imaging with a low-overlap, do you have to pre-train the network on highly overlapping data first? If that is the case, then the total dose incident onto the sample will be determined by both the high and low overlap data, making the dose saving smaller. Could you comment on the most optimal way to pre-train the neural network for low-dose imaging?

9. In the “Robustness in Low Light” section the authors say: “This approach has the advantage of not requiring retraining of the neural network, but only works with moderate scaling factors.”

Which means that you must re-train the neural network for different exposure times? The sample absorption also varies sample-to-sample, changing the photon number incident onto the detector. Does it mean that the network must be trained on every single sample used for an X-ray imaging experiment? Could you expand more the compatibility of the pre-trained networks?

10. In the “Robustness in Low Light” section the authors say: “The continual learning strategy is thus essential to achieve accurate AI inference on new features”

If training must be carried out on the whole dataset to reach maximum quality, it sounds to me as being computationally expensive. What is the training time in this case? How does it compare to iterative phase retrieval? This would convince the readers to consider a switch from iterative to neural phase retrieval.

11. In the conclusion section the authors say that high-resolution X-ray imaging can be performed using the neural network: “First, by performing AI inference on individual diffraction patterns, the oversampling constraint is eliminated. We show sample image with an accuracy of 90% without any probe overlap between the measured points. This reduces the beam dose by a factor of 6

compared to that required by conventional methods. In the second approach, the beam dose is further reduced by a factor of 10 to 1000 by performing AI inference on low counts data.”

I did not see proof of this in the paper. To achieve the desired resolution in ptychography, one must collect sufficient photons at the highest scattering angles incident onto the sensor. To collect the desired photon counts two methods can be used: short exposure/high-overlap scans or long-exposure/lower-overlap scans. In both cases, the total dose deposited onto the sample can be equivalent. You have shown that you can reduce the overlap OR the exposure time, but you cannot claim that you can do both at the same time since the data will have a significantly lower dose. On this note, the low-overlap experiments could have been done with a significantly higher exposure time, making the results better than they should have been. Your results would be more convincing if you would plot the total dose vs resolution/accuracy or if you would be more transparent by stating X-ray dose imparted onto the sample.

12. In the X-ray Ptychography Experiment section, I don't see details about the probe size, scanning parameters etc. I think they were mentioned somewhere in the paper, but it would be good to have everything described clearly in a single section.

REVIEWER COMMENTS

Reviewer comments are in black. Our response is in blue. The original version of the manuscript text is italicized (blue) with the revised text highlighted in red.

Reviewer #1 (Remarks to the Author):

This paper describes a workflow combining high performance computing (HPC), deep learning, real-time inference on edge devices, to improve ptychographic imaging at a beamline, resulting in improved phase retrieval at lower doses and overlaps than using conventional algorithms.

This is an important paper in many respects. It is one of the few papers that connects beamline microscopy to HPC systems in an automated manner that includes a loop from experimental results. The former (connecting HPC to microscopes) is not necessarily new, but the latter most certainly is. The authors achieve this via continual training of a neural network, and continual evaluation of the performance of the neural net compared to the training data, to determine whether a new round of training is necessary for satisfactory performance. As such, this paper's novelty to me is that it shows the benefits of an automated, computational workflow in concert with a controllable experiment, yielding tangibly better results than the current state of the art.

We thank the reviewer for their encouraging comments. Please find below responses to the reviewer's questions point-by-point.

I have just a few questions / comments that I hope the authors will be able to address:

1. To what extent are the trained models reusable? Can a model be trained on a single sample, then used on the same sample after a few weeks (given drift in parameters)?

Strictly speaking, the trained models are not reusable under a different experimental condition (e.g., with a different x-ray source or a different sample). However, the model has been shown to be quite tolerant to small drift in parameters. Below we show two cases of drift. In the first case, we assume the incoming intensity has changed. This can happen for many reasons, such as a decrease of electron current in the storage ring, or simply the monochromator drifting off its rocking curve. We use the same model to infer on experimental data with different count rate acquired on the same area (RC_Figure 1a – 1d). The stitched inference for when the count rate has varied by a factor of 4 (RC_Figure 1f) shows no visible difference compared to the expected result (RC_Figure 1e). When the count rate has varied by a factor of 16 (RC_Figure 1g), the predicted sample structure still looks correct, but the predicted level of phase is off by as much as ± 0.2 rad. When the count rate has varied by a factor of 40 (RC_Figure 1h), even the predicted structure is wrong, and the stitched inference is no longer usable. This shows that the trained model can still produce reliable phase inference even if the incoming intensity has drifted to 25% of its initial level. The predicted phase will still be usable as feedback for experimental steering even if the incoming intensity has drifted to 6% of its initial level. We have added a new paragraph in the **Discussion** section and a new supplemental Figure S5 to include these new results.

RC_Figure 1: (a-d) example experimental diffraction patterns acquired on the same area of the sample, at different count rates. (e-h) stitched phase inference from the same model on data acquired at different count rates. The count rate for the trained model is the same as the data shown in (a).

Next, we assume the focus of the x-ray beam has changed. This can happen after moving the x-ray beam on the sample over a large distance. If the direction of the movement is not perfectly parallel to the sample surface, and is at, for instance, an angle of 1° with regard to the sample surface, moving the x-ray beam by 1 mm on the sample surface will shift the focus by about 15 μm . The training data was prepared using experimental data acquired with an intentionally defocused beam (RC_Figure 2a). More specifically, the focusing optics was positioned at -210 μm from its focal point to create a large footprint of 800 nm FWHM on the sample. RC_Figure 2b and RC_Figure 2c shows respectively the probe amplitude for focusing optics positioned at -195 μm and -265 μm from its focal point. The beam size was respectively 700 nm and 1 μm FWHM. RC_Figure 2e and RC_Figure 2f show the corresponding stitched inference using data acquired with these two probes. For a shift in focus of +15 μm , the stitched inference (RC_Figure 2e) is similar to what was obtained with iterative phase retrieval (RC_Figure 2d). There is a small blurring in the phase structures, which indicates a slight decrease in spatial resolution. For a shift in focus of -55 μm , significant blurring was observed in the stitched inference (RC_Figure 2f). The spatial resolution was a lot worse, but the phase structure was largely correct. This shows that the trained model can still produce reliable phase inference if the focus has shifted by 15 μm from its nominal position. The predicted phase will still be usable as feedback for experimental steering if the focus has shifted by 55 μm from its nominal position.

RC_Figure 2: (a) probe amplitude used for training. The optics was positioned at -210 μm from its focal point to create a large footprint on the sample. Probe amplitude when the optics was positioned at (b) -195 μm and (c) -265 μm from its focal point. (d) iteratively retrieved phase serving as the groundtruth for comparison. (e) stitched inference result using data acquired with the probe illustrated in (b). (f) stitched inference result using data acquired with the probe illustrated in (c).

2. Is there any uncertainty quantification (even using something simple like dropout) that could be implemented for the algorithm? One of the concerns here is that if there is an underlying distributional shift, then lack of UQ can mask errors in the inference, so have some UQ would be advantageous.

The reviewer's comment on the distribution shift is an insightful and important comment and we thank them for raising this. We have explored the use of Monte Carlo (MC) Dropout to estimate the epistemic uncertainty in the model. RC_Figure 3 shows the iteratively reconstructed phase, the difference to the PtychoNN prediction (Phase Retrieval – PtychoNN) and the epistemic uncertainty as modeled by MC-Dropout.

RC_Figure 3: Iterative phase retrieval, difference between PtychoNN and phase retrieval and epistemic uncertainty estimated through MC-Dropout. Estimated uncertainty is often highest in locations of high gradient.

Perhaps expectedly, we observe the highest uncertainty at the boundaries of the features where there is a large phase gradient. However, we did not use MC-Dropout or other estimators for uncertainty like network ensembles or Bayesian NNs because of the additional overhead. For instance, evaluating uncertainty through MC-Dropout requires multiple forward passes through the network which would slow our throughput down by a factor of 10 or more (depending on how many forward passes).

We note that since we continue to perform phase retrieval, we use the phase retrieval results to monitor for distributional shift. As is elaborated in answers to the question below, this approach is accurate and easy to implement, but cannot produce real time results. In the future we hope to develop a workflow that incorporates two separate streams, the first through a model that we would use for fast training and inference, and the second for uncertainty quantification (both aleatoric and epistemic) for the real time monitoring of any distributional shift.

3. On the question of distributional shift, is this something that could occur during the experiment itself? How would one detect it if so and deal with it? (I can think of examples, such as ensemble learning or cycle-GANs that could be used to counter it).

We thank the reviewer for the intriguing question. The x-ray nanoprobe beamline at the Advanced Photon Source, where the experiment took place has a laser interferometry feedback system capable of maintaining positional accuracy to 5 nm. As such we don't expect much distributional shift caused by changes in the probe (illumination) function. On the other hand, changes in the sample itself, are completely expected. Because the workflow is designed to guide explorational experiments, it is highly likely for the x-ray beam to venture into an area quite different from those used to train the existing model. In our current design, such distributional shift is detected through comparison with the ground truth at regular intervals (for instance, once every hour). We have updated the discussion section to offer a clearer explanation as to how it works. The original version is italicized with the added text highlighted in red. As above, we add the qualifier that this is not a real-time strategy and there could be scenarios where the sample changes dramatically in the time it takes to obtain the phase retrieval results.

Because access to on-demand HPC resources may be limited, a strategy is devised to minimize the use of expensive HPC resources. In the initial phase, when a new set of experimental data is received at the HPC, it automatically triggers the image reconstruction service using iterative methods. The results are then cropped and paired with the corresponding diffraction patterns to form the labeled training data. Part of this training data is used immediately to validate the accuracy of the existing neural network. If a large mismatch is detected between the AI-inferred phase from the latest model and the iteratively retrieved phase in the training data, a retraining service is queued which upon completion also sends the updated model to the edge device. If the mismatch is within a certain tolerance level (for instance, a difference of less than 10% in structural similarity), the automatic image reconstruction and retraining services are suspended, with the experimental data being sent and reconstructed at HPC periodically at a prolonged interval (for instance, once every hour). Comparing

the AI inference using the existing neural network with the periodically retrieved phase is particularly useful to check, among others, data distributional shifts. The on-demand image reconstruction and retraining services are resumed if a large mismatch is again detected.

4. Is there an option to incorporate simulations into this workflow in the future?

We thank the reviewer for this great comment.

There are a few ways to incorporate simulations into the workflow, the most straightforward being using simulations to generate data used for the training of the NN[1]. The main advantage with this approach is that the workflow would be operational right from the start. However, such approach requires an accurate knowledge of the probe function in order to calculate the associated diffraction patterns. Any discrepancies in the probe function will naturally introduce a covariant data distributional shift in the system. It also requires an accurate digital twin of the sample, with the right chemical composition and thickness to produce the right refractive index, because ultimately what the NN learns is the mapping between a spatial distribution of refractive indices and an image of intensities. Finally, even with the correct probe function and sample digital twin, we find that training with simulated data is still less accurate than training with experimental data (current work). We attribute this to the inability of the simulations to account for real life imperfections in the data caused by beam incoherence, instrument vibrations, noise, etc. It was for these reasons that we had opted not to incorporate simulations in this workflow.

[1] British Machine Vision Conference 2019, 2019 278.

Reviewer #2 (Remarks to the Author):

In this article, Babu et al. apply a deep convolutional neural network (i.e., PtychoNN 2.0, modified from PtychoNN) for real-time ptychographic imaging. The presented Machine Learning (ML) approach demonstrated the possibility of the real-time ptychographic image. However, while the authors claim in the article that the iterative method is slow, the proposed ML approach still needs the iterative reconstruction method. This will limit the application of ML-based approach and also affect the general of the proposed ML approach.

We thank the reviewer for this comment.

While it is true that the proposed ML approach is still dependent on the slower iterative reconstruction method, such dependency is removed as soon as the training is complete. As long as no distributional shift is detected, the ML-based approach can independently achieve real-time phase retrieval for the remainder of the experiment.

In fact, we believe the dependence on iterative phase retrieval is what makes the proposed approach a more general solution to the real-time ptychographic imaging problem. For instance, it is possible to train the NN using synthetic, simulated data. But as we pointed out in the response to reviewer #1 comment #4, such approach is sample and probe specific, and is often less accurate. Because iterative phase retrieval can be applied on any samples under any illumination conditions, through dependence our proposed approach also becomes sample-probe agnostic.

Moreover, despite the interest in applying a modified deep neural network for real-time application, the discussion about the comparison between the iterative phase reconstruction and the ML-based method shows that the proposed ML approach gives a worse reconstruction compared to the iterative method. As shown in Fig. 2C, Fig. 3E, and Fig. 3F, the ML approach cannot recover the fine feature of the measured sample. Even there is a speed advantage for the ML-based approach, reconstruction accuracy should be more important than reconstruction speed.

Therefore, I would not recommend this work for publication in Nature Communication but in a more specialized journal focused on AI techniques, unless they can make a significant improvement in the performance of the ML-based approach over the iterative method.

We thank the reviewer for this comment.

It was never our intention to achieve higher accuracy with a ML-based method compared to numerical methods (such as iterative phase retrieval). In fact, it is a common practice to perform a few iterations of numerical optimization after AI-inference, because the former is known for its superior accuracy [1-3].

On the note of accuracy, we also respectfully disagree with the reviewer's claim about ML not recovering fine features in Fig. 2C, Fig. 3E and Fig. 3F. While it is true that the accuracy of the workflow would not exceed that of interactive phase retrieval, we do believe that the ML inference was able to reproduce all the fine details present in those figures.

Finally, during a high data rate synchrotron experiment, speed is just as important as accuracy. Scientific users would have all the time they need after the experimental beamtime to perform more accurate (and slower) data analysis, but they need real-time feedback during the experiment to steer the experiment in the right direction. This is particularly true for *in situ* experiments where samples undergo irreversible changes. The situation is even more challenging with the advent of diffraction limited storage rings (4th generation synchrotrons). The data acquisition speed is dramatically increased with the 100-fold increased brightness, which in turn requires the development of even faster real time data analysis workflows. We believe that this manuscript presents a significant advance by providing a solution to the problem of real-time analysis (and feedback), allowing the use of next generation to their fullest capacity. Please find below the response to the rest of the comments point-by-point.

[1] Henry Chan, Youssef S. G. Nashed, Saugat Kandel, Stephan O. Hruszkewycz, Subramanian K. R. S. Sankaranarayanan, Ross J. Harder, Mathew J. Cherukara; Rapid 3D nanoscale coherent imaging via physics-aware deep learning. Applied Physics Reviews 1 June 2021; 8 (2): 021407.

[2] Yao, Y., Chan, H., Sankaranarayanan, S. et al. AutoPhaseNN: unsupervised physics-aware deep learning of 3D nanoscale Bragg coherent diffraction imaging. npj Comput Mater 8, 124 (2022).

[3] Wu, L., Juhas, P., Yoo, S. & Robinson, I. (2021). Complex imaging of phase domains by deep neural networks. IUCrJ, 8, 12–21.

Below are some additional questions:

1. The detailed architecture of the ML model is missing in the article, please clarify it in the paper.

We thank the reviewer for this comment. We have clarified the architecture in the **Methods** section and added a new supplementary figure S7. As shown in RC_Figure 4, PtychoNN 2.0 is a fully convolutional autoencoder network with an encoder-decoder architecture. The convolution layers use a filter size of 3x3 and a stride of 1. Here “Conv” refers to 2D convolution followed by a ReLU activation function, “Pool” corresponds to a 2D maxpooling operation with a pool size of 2x2, and “upsampling” uses a 2D bilinear interpolation for a 2x2 input. The final “TanH” activation layer outputs predictions in the range [-1,1], which are then scaled to get output phases in the range [-pi, pi]. The output shapes in each block are read as [batch size, number of filters, height, width]. The number of trainable parameters is also indicated in parenthesis against each block in the figure.

RC_Figure 4: Detailed architecture of PtychoNN 2.0.

2. What is the main difference between PtychoNN 2.0 and PtychoNN?

We thank the reviewer for this insightful comment. We have added more clarifications in the **PtychoNN 2.0 Architecture** subsection under the **Methods** section. Below is an excerpt of the revised manuscript.

The detailed architecture of PtychoNN 2.0 is shown in supplementary Figure S7 and this network differs from PtychoNN in two major aspects. First, it predicts only the phases and therefore does not contain the amplitude branch originally present in PtychoNN. Second, each convolutional layer in PtychoNN 2.0 contains 0.5x the number of filters as PtychoNN. With these modifications, the number of trainable parameters in PtychoNN 2.0 is reduced to 0.7M parameters compared to 4.7M parameters in PtychoNN. Reducing the model size has the advantage of improving the training times and helps in lowering the inference latency at the edge without significantly impacting the accuracy. The performance of the two ML models is analyzed in terms of the quality of the phase predictions and inference times. Supplementary Figure S8 indicates that PtychoNN 2.0 can give equally good predictions as PtychoNN.

3. What is the reconstruction resolution for Figure 2, Figure 3, and Figure 4? The structural similarity can be used only when the ground truth is known. For ptychography, the ground truth is generally not known. As a widely adopted approach, the authors should use the Fourier shell ring method instead of structural similarity.

We thank the referee for their suggestion. To address the reviewer's comment, we have picked two scans that share a large overlap area. We note that despite covering the same area, the beam positions on the sample for these two scans were completely different.

RC_Figure 5 shows the iteratively reconstructed phase of the two scans, with the overlap area highlighted by the red rectangle. We have performed Fourier shell correlation calculation on the overlapped area, the resolution of iterative phase retrieval was about 35 nm, as indicated by the image below. The beam size on the sample was about 800 nm for those measurements.

RC_Figure 5: iteratively retrieved phase for scan frc1 and scan frc2. The red rectangle marks the common area where we have performed Fourier shell correlation calculations.

Next, we perform NN inference on these two scans. For this test, we retrained the NN after specifically removing those two scans from the training data. The stitched inferences for the two scans are shown in RC_Figure 6, with the overlap area highlighted in red. Fourier shell correlation performed on the overlap area of the two stitched inference images indicates a resolution of 20 nm, much better than the number obtained for iterative phase retrieval. We are not confident this reflects the actual resolution of the NN inference and have hence chosen to retain SSIM as the metric of choice. However, we have included these results in the **Methods** section and as Supplemental Figure 11.

RC_Figure 6 : stitched inference for scan frc1 and scan frc2. The red rectangle marks the common area where we have performed Fourier shell correlation calculations.

4. When the different counts of diffraction data from the same region are given to the model, will the ML model give the same results or different results?

We thank the referee for this insightful comment. The NN maps intensity levels to phase information. For this reason, when given drastically different counts of diffraction data from the same region, the ML model is expected to produce different results. However, the model has been shown to be quite tolerant to small changes in the count rate of the diffraction data. For the following, we use the same model to directly infer on experimental data with different count rate acquired on the same area (RC_Figure 7a – 7d). The stitched inference for when the count rate has varied by a factor of 4 (RC_Figure 7f) shows no visible difference compared to the expected result (RC_Figure 7e). When the count rate has varied by a factor of

16 (RC_Figure 7g), the predicted sample structure still looks correct, but the predicted level of phase is off by as much as ± 0.2 rad. When the count rate has varied by a factor of 40 (RC_Figure 7h), even the predicted structure is wrong, and the stitched inference is no longer usable. This shows that the trained model can still produce reliable phase inference even if the count rate has varied by as much as a factor of 4. We have added a new paragraph in the **Discussion** section and a new supplemental Figure (S5) to include these new results. To deal with more drastic variations in count rate, we have described two strategies that can be found in the **Robustness in Low Light** section.

RC_Figure 7: (a-d) example experimental diffraction patterns acquired on the same area of the sample, at different count rates. (e-h) stitched phase inference from the same model on data acquired at different count rates. The count rate for the trained model is the same as the data shown in (a).

5. How is the stitching of individual reconstructed images from the ML model performed? Please give the details.

A new sub-section has been added to the **Methods** section to specifically clarify this procedure.

Stitching of AI inference results

The workflow produces, for each input diffraction pattern, an inference image of 128x128 pixels. Each inference image is centered around the corresponding beam position on the sample which, for spiral scans, is not on a regular grid. The first step of stitching is thus to interpolate all the AI inference results onto the same regular grid, based on information such as the beam position and the pixel size of the inference image. The stitched image is calculated by statistically averaging all the interpolated results on the same but larger regular grid. Alternatively, we have performed weighted average of the interpolated results using the probe amplitude as the weights. The second approach yielded very comparable results as the original one with no visible improvement on the quality of the final image.

6. The image size for all the Figures in the paper are missing.

We apologize for this oversight. A scale bar has been added to all the real-space images.

7. When the focus of the used X-ray is changed or off, will the ML model give the same results?

We have performed additional tests to answer the reviewer's question. The NN is trained for a defocused probe with a FWHM of about 800 nm (see RF_Figure 8a). Below we are showing the stitched inference result for data acquired with a probe size of respectively 700 (RF_Figure 8b) and 1000 nm (RF_Figure 8c), achieved by moving the focusing optics along the optical axis by respectively +15 and -55 μm from its usual position. The stitched inference for the 700 nm probe (RF_Figure 8e) looks similar to the phase reconstructed with iterative methods (RF_Figure 8d). The stitched inference for the 1000 nm probe (RF_Figure 8f) appears to be blurry, but still retains all the coarse features of the sample. This shows that the ML inference would still function even when the focus of the used X-ray is off, at the expense of a degraded spatial resolution.

RC_Figure 8: (a) probe amplitude used for training. The optics was positioned at -210 μm from its focal point to create a large footprint on the sample. Probe amplitude when the optics was positioned at (b) -195 μm and (c) -265 μm from its focal point. (d) iteratively retrieved phase. (e) stitched inference result using data acquired with the probe illustrated in (b). (f) stitched inference result using data acquired with the probe illustrated in (c).

We note that the workflow is capable of detecting changes in the x-ray focus itself. This is achieved through the same process as the detection of data distributional shift. When the focus has drifted, a low structural similarity value is obtained between the AI inference and the iteratively retrieved phase. This triggers a retraining of the neural network using the latest data, which consists of primarily data acquired with the drifted probe. As the experiment continues, more data with the drifted probe are collected, and converted to training data, eventually getting the inference accuracy back up through the use of continual learning. Below we have attached the paragraph from the **Discussion** section that addresses this issue. The revisions are highlighted in red.

Because access to on-demand HPC resources may be limited, a strategy is devised to minimize the use of expensive HPC resources. In the initial phase, when a new set of experimental data is received at the HPC, it automatically triggers the image reconstruction service using iterative methods. The results are then cropped and paired with the corresponding diffraction patterns to form the labeled training data. Part of this training data is used immediately to validate the accuracy of the existing neural network. If a large mismatch is detected between the AI-inferred phase from the latest model and the iteratively retrieved phase in the training data, a retraining service is queued which upon completion also sends the updated model to the edge device. If the mismatch is within a certain tolerance level (for instance, a difference of less than 10% in structural similarity), the automatic image reconstruction and retraining services are suspended, with the experimental data being sent and reconstructed at HPC *periodically* at a prolonged interval (for instance, once every hour). *Comparing the AI inference using the existing neural network with the periodically retrieved phase is particularly useful to check, among others, data distributional shifts.* The on-demand image reconstruction and retraining services are resumed if a large mismatch is again detected.

8. Did the author try the Difference Map (DM) method for these experimental data? To make a fair comparison, the author should also use the DM iterative method.

We thank for the reviewer for this comment., all the iterative retrieved phases shown previously in this manuscript were obtained with the rPIE algorithm.

To answer the reviewer's question, we have performed additional phase retrieval with the DM method implemented in the latest stable release of Tike (v 0.24.2) [1], to compare with those retrieved with rPIE shown in the main manuscript. As shown in RC_Figure 9. The phase retrieved with both methods are mostly consistent with each other. The phase retrieved with DM is slightly less uniform, which is why we would still prefer to keep the rPIE retrieved phase for this paper.

All the reconstructions shown in RC_Figure 9 were performed on a RTX 2080 Ti GPU, with 3 probe modes and 2000 iterations. This result has been added to the **Methods** section under **Iterative Phase Retrieval and Image Processing Toolkit**, and as Supplemental Figure S6.

RC_Figure 9: (a) DM and (b) rPIE retrieved phase for data shown in main manuscript Figure 2B. (c) DM and (d) rPIE retrieved phase for data shown in main manuscript Figure 5B.

[1] Tik repository for ptychography software <https://github.com/AdvancedPhotonSource/tike/releases/tag/v0.24.2>

9. When the training data size is increasing, will the model need a longer time to finish online training?

We thank the reviewer for this insightful question. We have performed a study on the training time as a function of data size. The result is shown in RC_Figure 10. The training time varies almost linearly with the size of the training data. For data specific to this work, one needs less than 20000 data pairs to reach a reasonable accuracy (>90%, see Fig. 5 of the original manuscript) in the inference results, and it takes about 4 min to train using 8 A100 GPUs. To train on the entire dataset (~100000 data pairs) collected during this experiment takes about 10 min. Further speed-up of the training time can be achieved by

spreading the NN training over more GPU nodes at the HPC. For the live demonstration, for example, we had reserved 64 A100 GPUs at the ALCF.

The Neural Network Architecture and Training subsection has been revised according to include:

The training time scales almost linearly with the size of the training data. For 100,000 pairs of data, the training time is about 10 min using 8 A100 GPUs at the ALCF.

RC_Figure 10 has been added as Supplemental Figure S10.

RC_Figure 10 : Training time versus training data size on 8 A100 GPUs.

10. What method is used to avoid the overfitting of the ML model? Why did you choose 90% of the data as training and 10% as validation? Did you have separate images for testing or your validation and test are the same?

We thank the reviewer for this question. The training set used in the experiment consisted of ~100000 data pairs and for every epoch we randomly selected 10% of the training data as the validation. The choice of 90/10 seems reasonable in our experiment due to the large training dataset. The evolution of training and validation error over 50 epochs are shown in the RC_Figure 11. At each stage we run the training loop for 50 epochs, then take the network weights from the epoch where the validation loss is lowest [1]. During the inference, the neural network will infer on previously unseen test images that are not present in training or the validation datasets.

RC_Figure 11: **A** Cyclic learning rate policy adopted for the training with an initial value of 0.0001. The maximum and minimum learning rates are 0.001, 0.0001 respectively. **B** Variation in training and validation loss over 50 epochs estimated at two different iterations of the continual learning approach when ptychography scan #1 and scan #111 were added to the existing training data.

We have made the following changes to the **Neural Network Training** subsection under the **Methods** section. RC_Figure 11 has been added as Supplemental Figure S9.

Network weights are updated using the adaptive moment estimation (ADAM) optimizer to minimize the mean absolute error (MAE) between the target labels (output of iterative phase retrieval) and AI inferences. The model state at each training epoch is evaluated on unseen validation data, which is 10% of the training data. The validation data set is randomly chosen from the training data for every epoch. The trained model with the lowest validation loss at the end of 50 epochs is pushed to the edge device for the AI inference on subsequent scans. During the inference, the neural network will infer on unseen test images that are not present in the training or the validation datasets. Supplemental Figure S9 shows the training and validation losses for the first training iteration with only one ptychographic dataset (scan #1), and for training iteration 111 with 111 ptychographic datasets (when scan #111 was added). While we do not see evidence of overfitting in the early iterations, we notice that the training loss slightly diverges in the latter iterations. Here, the cyclic learning rate approach was used at each training iteration to ensure that we are not stuck in a local minimum even as we add more training data.

[1] Prechelt, L. (1998). *Neural Networks*, 11(4), 761–767.

Reviewer #3 (Remarks to the Author):

The authors present a neural-network framework for live ptychographic reconstruction feedback. The proposed framework claims that it can eliminate the overlap constraint in ptychography and enable low-dose imaging. The claims are supported by experimental results where the data was acquired at an impressive rate of up to 2kHz. The results show that the trained neural network reconstruction accuracy is as good as that of iterative phase retrieval, once all training data is taken into account. The neural network can also be pre-trained on different sample types (that meet specific criteria) which can be useful for practical experiments. There are also results showing the performance of the neural network for low-overlap scanning and low-exposure imaging. I believe this work has great potential for the 4th generation

synchrotron beamlines where the radiation dose and the resulting sample damage will be ultimately limiting the resolution at which the samples can be imaged. Especially for ptychographic X-ray tomography. The prospect of low-overlap and low-exposure ptychography sounds very promising for low-dose, high-speed ptychographic imaging. The neural network is also a great candidate for high-speed data-processing which will only increase in the next generation X-ray sources.

We thank the reviewer for their encouraging comments. Please find below response to the reviewer's comments point-by-point.

However, there are several general remarks that still remain upon reading the manuscript: 1. While the introduced workflow is claimed to be very fast, I lacked information on how long it takes to train the neural network and how long it takes to reconstruct the data using a neural network. I would also like to see a comparison between the neural network training speed and iterative phase retrieval.

We thank for the reviewer for pointing this out.

As shown in RC_Figure 12, the training time scales with size of the training data. For data specific to this work, one needs about 10000 data pairs to reach a reasonable accuracy (>90%, see Fig. 5 of the original manuscript) in the inference results, and it takes about 3 min to train using 8 A100 GPUs. To train on the entire dataset (100000 data pairs) collected during this experiment takes about 10 min. Further speed-up of the training time can be achieved by spreading the NN training over more GPUs nodes at the HPC. For the live demonstration, for example, we had reserved 64 A100 GPUs at the ALCF.

The Neural Network Architecture and Training subsection has been revised to include the following:

The training time scales almost linearly with the size of the training data. For 100,000 pairs of data, the training time is about 10 min using 8 A100 GPUs at the ALCF.

RC_Figure 12 has been added as Supplemental Figure S10.

RC_Figure 12 : Training time versus training data size on 8 A100 GPUs.

Once trained, the phase retrieval speed of the NN depends entirely on the inference time, which in turn depends on the machine we use. The actual inference time was separately shown in Table 1 and Table 2 of the **Methods** section in the original manuscript. For an edge device with small form factor, the inference time is about 2.3 ± 0.4 ms per image, corresponding to a frame rate of roughly 500 Hz. To achieve the 2 kHz inference claimed in the latter part of the paper, we had to use a more powerful machine with a RTX

A6000 GPU. With that, the inference time was reduced to 70 ± 5 us per image, corresponding to a maximum frame rate of 14 KHz.

Finally, the phase retrieval time for conventional methods depends on the number of iterations. It took about 1 min per 500 iterations for a scan consisting of 963 images on a machine with 8 A100 GPUs. This corresponds roughly to a frame rate of 16 fps. As such, the NN inference is more than 800 times faster than conventional methods when fully trained.

We have revised The **Discussion** section include the following comparison.

*Using a powerful GPU, the inference time was further reduced to 70 us per image (see **Methods**), corresponding to a frame-rate of 14 kHz. For comparison, 500 iterations of iterative phase retrieval took about 1 min for a scan consisting of 963 images, corresponding to an equivalent frame-rate of about 1000 times slower, at 16 Hz.*

2. The authors could also be more transparent on the capabilities of the neural network. My impression was that the two imaged samples had the same refractive index. As I read the paper I discovered that there are significant constraints on the neural network pre-training such as the need to use samples with the same refractive index, the need for the same exposure times etc. Having a discussion on the practical capabilities of the pre-trained neural network would be useful to the reader.

We thank for the reviewer for this insightful comment.

We have added a new paragraph in the **Discussion** section to specifically discuss the limitations and tolerance of the neural network.

One of the reasons behind the high inference accuracy demonstrated in this work is that the neural network is trained specifically for samples with a given range of refractive indices and for a fixed illumination probe. This significantly simplifies the intensity (-input) phase (-output) relationship for the neural network, thus allowing it to accurately solve the inverse problem. As long as the refractive index of the sample remains in the same range as those used in the training data, the neural network can accurately predict new features it has never seen such as those demonstrated in Figure 2. If the refractive index of the sample falls outside of the range however, a retraining of the neural network is recommended. We note that it is not necessary to retrain neural network for small fluctuations in the sample or in the probe. In fact, the neural network has been shown to be quite tolerant to those changes. In Supplemental Figure S5, we use the same model to directly infer on experimentally acquired data with different count rate from the same area. The stitched inference for when the count rate has varied by a factor of 4 shows no visible difference compared to the expected result. When the count rate has varied by a factor of 16, the predicted sample structure still looks correct, but the predicted level of phase is off by as much as ± 0.2 rad. When the count rate has varied by a factor of 40, even the predicted structure is wrong, and the stitched inference is no longer usable. This shows that the trained model can still produce reliable phase inference even if the count rate has varied by as much as a factor of 4. The predicted phase will still be usable as feedback for experimental steering even if the incoming intensity has varied by as much as a factor of 16.

3. The authors could also improve the presented results and claims of low-dose imaging using the neural network. For example, the neural network was trained on highly overlapping data, which was then used to reconstruct low-overlap data of the same sample field-of-view. Does the neural network have to be pre-trained on highly overlapping data? Is the accuracy of the pre-trained model maintained when imaging a different sample field-of-view with a reduced overlap? For example, to image a $1 \times 1 \text{ mm}^2$ area, can you image a micron sized area with high overlap and then image the remaining area with a virtually zero overlap? Addressing these comments would either improve the impact of the paper and/or inform the reader on the use-case of the proposed workflow.

We thank for reviewer for this comment.

No, the neural network does not need to be pre-trained on highly overlapped data. The reason for using highly overlapped data is because they are needed by iterative phase retrieval, which provides the training data for the neural network.

To answer the follow-up question. Yes, the accuracy of the pre-trained model is maintained when imaging a different sample field-of-view with a reduced overlap. We apologize if we had not made it clear in the main manuscript. All the data points shown in Figure 3A were obtained after stitching together NN inferences at various reduced overlap (from 0 to 90%). These stitched results are shown in RF_Figure 13. We have also added them to the revised manuscript as Supplemental Figure 2.

It is obvious from RF_Figure 13 that the accuracy of the NN inference is independent of the overlap ratio. We note that some errors can be spotted in the stitched inference, which manifest themselves as sharp vertical or horizontal lines. These errors were not caused by inaccuracy in the NN inference, but rather by the way we generate these “low overlap scans”. To generate a scan with an overlap ratio of 0.1 for instance, we selectively remove data points from one parent scan which has an initial overlap of 0.6, until the average overlap ratio is 0.1. Because the original scan was measured with an irregular spiral pattern, it is hard to achieve a homogeneous overlap of 0.1 everywhere in the new data. Instead, some areas may have an overlap of 0.2, while some others may have a negative overlap (i.e. the step size is larger than the probe size). It is those areas with a negative overlap ratio where aforementioned errors can be spotted, because none of the stitched inference can produce reliable phase prediction on the non-overlapped area.

RC_Figure 13: Stitched NN inference at various reduced overlap. Results of iterative phase retrieval is also shown for comparison. To create scans with overlap ratio between 0 and 0.5, we selectively remove data points from a parent scan with an initial overlap of 0.6 until the average overlap matches the desired value. To create scans with overlap ratio between 0.6 and 0.8, we selectively remove data points from another parent scan with an initial overlap of 0.9 until the average overlap matches the desired value.

To answer the final question. Yes, it is possible to image a micron sized area with high overlap, then image the remaining area with virtually zero overlap, as long as the micron sized area contains enough diversity. While we have not taken the necessary data to demonstrate this, we are confident of this statement. As pointed out previously, the NN is simply unaware of the concept of spatial overlap.

4. There were also claims of low-dose imaging, but the authors never really stated the actual dose imparted onto the sample. While there are two sections describing low-exposure imaging and low-overlap imaging, in each of the cases the total dose incident onto the sample can be misleading to the reader. For example, low-exposure imaging with an overlap ratio of 99% can have the same dose as an experiment with a moderate exposure time and a moderate overlap. Since dose is linked to both the exposure time and the overlap percentage more details are necessary to assess the importance of the proposed results for low-dose experiments.

We thank the reviewer for this comment.

We apologize for not disclosing the actual dose. It was not intentional. There was initially a subplot in Figure 4 that showed the calculated dose as a function of exposure time, which was removed in the final draft for simplicity.

We have updated Fig. 3 in the **Low-dose Ptychographic Imaging through Sparse-sampling** section to show the effect of overlap in actual photon counts. We have attached the revised version of Fig. 3 below (RC_Figure 14). The average dose is 20k ph/nm² for an overlap ratio of 0.9, 1600 ph/nm² for an overlap ratio of 0.6 and 250 ph/nm² for an overlap ratio of 0. These values are also listed in the revised figure caption. To achieve the same accuracy of 0.8, iterative methods requires an overlap ratio of 0.6 while our proposed method works with an overlap ratio of 0. In the original manuscript we have concluded that this allows us to lower the dose by a factor of 6.25 using the squared ratio of the step size (460 nm vs 180 nm). Here, the same factor can be confirmed by directly comparing the average dose as $1600/250 = 6.4$.

RC_Figure 14: updated Fig. 3B in the revised manuscript. **A** shows the accuracy of both AI inference and iterative phase retrieval, as a function of the overlap ratio. Also shown are visualizations of the actual probe overlap for overlap ratios of **B** 0.9 **C** 0.6, and **D** 0, by plotting the photon dose at each scan positions. The average dose is 20k ph/nm² for an overlap ratio of 0.9, 1600 ph/nm² for an overlap ratio of 0.6 and 250 ph/nm² for an overlap ratio of 0. A defocused probe is used, resulting in a donut-shaped illumination on the sample that can be individually discerned at low overlap.

The reviewer is also correct that low-exposure imaging with an overlap ratio of 99% can have the same dose as an experiment with a moderate exposure time and a moderate overlap. To clarify this, we have added the estimated dose in the **Robustness in Low Light** section. Below is an excerpt of the revised manuscript. The changes are highlighted in red.

A video recording of the live demonstration of inference at 2kHz can be found here. The estimated dose for this demonstration was 6 ph/nm².

Below you can find specific comments to parts of the manuscript:

5. In the Introduction you say: “The inversion of ptychographic data (or phase retrieval) provides a solution to the phase problem, where only the amplitude information about the sample, and not its phase, is retained in the measured intensities.” This sentence seems to suggest that the intensity of the measured diffraction patterns encodes the absorption of the sample, while the phase of the diffraction patterns encodes the phase of the sample. This is not true since the measured intensities (diffraction patterns) contain information of both the absorption and phase of the sample. Otherwise, phase only samples would have zero diffraction signal.

We thank for the reviewer for pointing this out. We have replaced the misleading piece in the introduction with the following description.

The inversion of ptychographic data (or phase retrieval) provides a solution to the phase problem, where only the amplitude information about the wave exiting the sample, and not its phase, is retained in the measured intensities.

6. In the Introduction you say: “Currently, real-time imaging with ptychography is limited by the use of conventional phase retrieval methods. These methods do not produce live results until after acquiring a few tens if not hundreds of diffraction patterns, despite substantial advancement in the reconstruction algorithms.”

Could you clarify what you mean by real-time imaging? There are X-ray synchrotron beamlines that can reconstruct ptychographic data faster than the data acquisition rate (acquisition taking up to a minute for high-resolution measurements). I would argue that current existing algorithms are fast enough for ptychographic tomography, so it depends what you mean by real-time imaging. Are you referring specifically to focused-beam ptychography? For example, near-field ptychography uses a huge probe and can capture only 10-20 diffraction patterns to cover a wide sample area where the reconstruction speed is limited not by iterative phase retrieval but rather by the I/O operations of the data. Also the number of diffraction patterns is proportional to the sample field-of-view, so I don't really see the link with reconstruction algorithm advancements. If you want to reconstruct a meaningful field-of-view, the 10-100 diffraction patterns you are referring to must be captured for visualization to happen. I think you should emphasize more the super high speed imaging at 2kHz that you are doing where the conventional phasing algorithms are definitely too slow.

We thank the reviewer for their suggestion.

In our opinion, there are two aspects to the term “real-time imaging”. 1 How soon can the algorithm show the reconstruction results, and 2 how fast can it update them.

Regarding aspect #1: Conventional iterative phase retrieval methods require a few tens of images at least to be able to converge to a reasonable solution. For instance, it is not enough to just give three diffraction images to an iterative solver and expect an accurate reconstruction. This is what we meant when we wrote *These methods do not produce live results until after acquiring a few tens if not hundreds of diffraction patterns*. In comparison, the proposed workflow can show live image of the sample starting with just one single diffraction image.

Regarding aspect #2: It is true that there are advanced algorithms that can reconstruct faster than the data acquisition rate. However, they are still a lot slower than the maximum data acquisition speed allowed by state-of-the-art detectors and diffraction-limited x-ray sources. In Figure 6 of the original manuscript we show a factor of almost 3000 between the data rate of the latest high throughput phase retrieval algorithm [1,2] and the maximum data rate allowed by the latest generation of area detector [3]. With the 800x gain in phase retrieval speed demonstrated with our workflow (see response to comment #1), we believe we are one step closer to achieving “real-time imaging” even with the maximum data acquisition speed allowed by the latest generation of detector technologies.

[1] X. Yu, et al., *Scientific Reports* 12, 5334 (2022)

[2] V. Favre-Nicolin, et al., *Journal of Applied Crystallography* 53, 1404 (2020).

[3] I. Johnson, et al., *Journal of Instrumentation* 9, C05032 (2014).

7. In the “Real-time Streaming Ptychography Imaging Workflow” section the authors say: “Iterative phase retrieval is typically too slow for low-latency feedback even if the processing is performed using HPC resources.”

I will disagree since this low-latency feedback is done routinely in beamlines currently without deep-learning phase retrieval methods. Define your criteria: 4th generation sources, low exposure imaging, 2kHz data collection etc.

We thank the reviewer for pointing out this lack of clarity.

We have revised the manuscript to be more specific about our criteria.

Original manuscript:

Iterative phase retrieval is typically too slow for low-latency feedback even if the processing is performed using HPC resources.

Revised manuscript:

While iterative phase retrieval has also been used to produce real-time feedback for ptychographic experiments, they are typically too slow for data acquired with state-of-the-art detectors running at their maximum speed (>1 kHz), which is going to be a common practice at the fourth generation synchrotrons.

8. In the “Real-time Streaming Ptychography Imaging Workflow” section the authors say: “Instead, the results of iterative phase retrieval are cropped and paired with the corresponding diffraction patterns to provide labeled data for incremental training of a neural network.” and “Hereinafter, one set of training data refers to one diffraction pattern paired with a cropped image of the iteratively retrieved phase”

Typically the diffraction pattern at a given scan point represents a certain field-of-view of the sample. By cropping, do you mean that you select a field-of-view area of the sample defined by the probe size and obtain a pair of field-of-view plus a diffraction pattern? Also when do you retrieve the recovered phase? After 1 iteration or more? Moreover, do you send the paired data for training after every single scan? More details would be helpful.

We thank the reviewer for this comment.

Indeed, each diffraction pattern represents a certain field-of-view of the sample illuminated by the probe at that scan position. We select this field-of-view based on the FWHM of the probe. For a reconstructed pixel size of 6.82 nm and a probe FWHM of 800 nm, an area of 128x128 pixels was cropped each time to be paired with the corresponding diffraction pattern for the training. Because the measurements were performed with a spiral scan on an irregular scan grid, instead of grabbing data directly from the iteratively retrieved result, the phase in the training data were interpolated on a regular grid with 128x128 pixels centered around the center of the beam illumination. This information has since been updated in the revised manuscript under **Neural Network Training** of the **Methods** section.

If by “when do you retrieve the recovered phase”, the reviewer is referring to the iteratively recovered phase, we typically use 500 iterations of rPIE before converting the result into training data.

Each scan contains 963 diffraction images. Initially, 963 pairs of training data were sent after every single scan. Once the inference accuracy is above a user-defined threshold (in our case, 90% of structural similarity compared to iteratively retrieved phase), the training process is paused. The quality of the inference at a prolonged delay (in our case, every hour), until low structural similarity values were detected. At which point, continual learning is resumed, and training data will be sent again after every single scan.

9. In the “AI Inference Accuracy at the Edge” section the authors say: “Comparison of the line profiles in Figure 2C shows that, once trained, AI inference results are almost identical to those obtained with iterative phase retrieval.”

Image quality depends on iteration number. Could you also state how fast the AI reconstruction is compared to the iterative engine in this case? Quality vs speed is an important metric.

We thank the reviewer for this comment. Indeed, quality vs speed is an important metric that we should carefully evaluate. We have addressed this problem in response to comment #1. Below is a short summary.

Assuming that 500 iterations is required for iterative methods to produce reasonable quality reconstructions, at the maximum frame rate of 14 KHz reported in this work the NN inference is about 1000 times faster than conventional methods which has an equivalent frame-rate of 16 Hz.

The **Discussion** section has been revised to include the following.

*Using a powerful GPU, the inference time was further reduced to 70 us per image (see **Methods**), corresponding to a frame-rate of 14 kHz. For comparison, 500 iterations of iterative phase retrieval took about 1 min for a scan consisting of 963 images, corresponding to an equivalent frame-rate of about 1000 times slower, at 16 Hz.*

10. In the “AI Inference Accuracy at the Edge” section the authors say: “Additionally, we note that the scanned area has features (alphabetical letters) that were not present in the training dataset.

This is the first mention that the training does not have to be done for each individual sample. Instead, the network can be pre-trained and re-used on other sample types. It would be useful to make this clear in the introduction of the technique.

As the reviewer has noted in their other comments, to accurately predict unseen features, the refractive index of the new features needs to be in the same range as those used in the training data. We have added a sentence in the **Discussion** section to clarify this argument.

As long as the refractive index of the sample remains in the same range as those used in the training data, the neural network can accurately predict new features it has never seen such as those demonstrated in Figure 2.

11. In the “AI Inference Accuracy at the Edge” section the authors say: “Supplementary figure S2 shows an example of the data included in the online training of the neural network which consists of entirely random patterns, although their refractive indices remain the same as the training and test data merely correspond to different regions of the same sample etched into different features.”

As I keep reading the text, I keep discovering where the neural network would and would not work. It would be useful for the reader to have a summary of what the neural network can do. For example, the pre-trained neural network can be used only on samples with the same-refractive index (as you state in this paragraph), the data must have the same exposure time (stated in the following paragraphs) etc.

We thank for the reviewer for this comment.

We have added a new paragraph in the **Discussion** section to specifically discuss the limitations and tolerance of the neural network.

One of the reasons behind the high inference accuracy demonstrated in this work is that the neural network is trained specifically for samples with a given range of refractive indices and for a fixed illumination probe. This significantly simplifies the intensity (-input) phase (-output) relationship for the neural network, thus allowing it to accurately solve the inverse problem. As long as the refractive index of the sample remains in the same range as those used in the training data, the neural network can accurately predict new features it has never seen such as those demonstrated in Figure 2. If the refractive index of the sample falls outside of the range however, a retraining of the neural network is recommended. We note that it is not necessary to retrain neural network for small fluctuations in the sample or in the probe. In fact, the neural network has been shown to be quite tolerant to those changes. In Supplemental Figure S5, we use the same model to directly infer on experimentally acquired data with different count rate from the same area. The stitched inference for when the count rate has varied by a factor of 4 shows no visible difference compared to the expected result. When the count rate has varied by a factor of 16, the predicted sample structure still looks correct, but the predicted level of phase is off by as much as ± 0.2 rad. When the count rate has varied by a factor of 40, even the predicted structure is wrong, and the stitched inference is no longer usable. This shows that the trained model can still produce reliable phase inference even if the count rate has varied by as much as a factor of 4. The predicted phase will still be usable as feedback for experimental steering even if the incoming intensity has varied by as much as a factor of 16.

12. In the “Low-dose Ptychographic Imaging through Sparse-sampling” section the authors say: “A spiral scan with an overlap ratio of 0.88 is used as the starting point, the iteratively retrieved phase of which serves as the ground truth. We then gradually reduce the overlap by selectively removing part of the data. At each step, a new iterative phase retrieval is performed alongside stitching of the AI inference results.”

As the overlap is decreased, you keep training the same network on the same sample. Since the neural network already knows how to do the phase retrieval of the highly overlapping data it seems to me that the phase retrieval task of the data with low overlap becomes much easier, hence the high-accuracy results. For imaging with a low-overlap, do you have to pre-train the network on highly overlapping data first? If that is the case, then the total dose incident onto the sample will be determined by both the high and low overlap data, making the dose saving smaller. Could you comment on the most optimal way to pre-train the neural network for low-dose imaging?

We thank the reviewer for pointing this out.

As mentioned in response to reviewer comment #3, the neural network does not need to be pre-trained on highly overlapped data. The reason for using highly overlapped data is because they are needed by iterative phase retrieval, which provides the training data for the neural network.

Because the neural network predicts phase on a per-diffraction-image basis, it was not trained to understand the concept of overlap, and was not aware of sample positions. Instead, an automated

program stitches individual inference produced by the neural network to form a complete picture of the sample based on its knowledge of the beam position for each data point.

In comment #3, the reviewer has suggested training using high overlap data on a small area, then image the remaining area with essentially no overlap. This is in our opinion the optimal way for low-dose imaging as it takes full advantage of the strengths of the proposed workflow. We thank the reviewer for this suggestion and we have added one sentence at the end of the **Discussion** section to describe this strategy.

Finally, we note that a good strategy for imaging beam-sensitive samples is to first run conventional ptychographic measurements in a small area with high overlap and decent count rate. This allows high quality training data to be acquired, which ensures the accuracy of the subsequent low dose imaging on the remaining area with essentially no overlap.

13. In the “Robustness in Low Light” section the authors say: “This approach has the advantage of not requiring retraining of the neural network, but only works with moderate scaling factors.”

Which means that you must re-train the neural network for different exposure times? The sample absorption also varies sample-to-sample, changing the photon number incident onto the detector. Does it mean that the network must be trained on every single sample used for an X-ray imaging experiment? Could you expand more the compatibility of the pre-trained networks?

We thank the reviewer for this comment. We shall address it in two parts.

If the sample has the same range of refractive indices as those used in the training data, but the number of incident photon has changed due to for instance a different exposure time or variations in synchrotron storage ring current, there are strategies that does not involve retraining of the neural network as described **Robustness in Low Light** section. Essentially this involves normalizing the inference input (diffraction pattern) to the same count rate as the training data before sending it to the NN. However, even without this normalization, the NN has been shown to be quite tolerant to small changes in the count rate of the diffraction data. For the following, we use the same model to directly infer on experimentally acquired data with different count rate from the same area (RC_Figure 15a – 15d). The stitched inference for when the count rate has varied by a factor of 4 (RC_Figure 15f) shows no visible difference compared to the expected result (RC_Figure 15e). When the count rate has varied by a factor of 16 (RC_Figure 15g), the predicted sample structure still looks correct, but the predicted level of phase is off by as much as ± 0.2 rad. When the count rate has varied by a factor of 40 (RC_Figure 15h), even the predicted structure is wrong, and the stitched inference is no longer usable. This shows that the trained model can still produce reliable phase inference even if the count rate has varied by as much as a factor of 4. The predicted phase will still be useable as feedback for experimental steering even if the incoming intensity has varied by as much as a factor of 16.

RC_Figure 15: (a-d) example experimental diffraction patterns acquired on the same area of the sample, at different count rates. (e-h) stitched phase inference from the same model on data acquired at different count rates. The count rate for the trained model is the same as the data shown in (a).

If the refractive index of the sample has changed dramatically, then it should be considered as a new experiment, and the NN has to be retrained. This issue was addressed in the **Discussion** session of the revised manuscript.

One of the reasons behind the high inference accuracy demonstrated in this work is that the ML model is trained specifically for samples with a fixed range of refractive indices and for a fixed illumination probe. This simplifies significantly the intensity (-input) phase (-output) relationship for the neural network, thus allowing it to accurately solve the inverse problem. As long as the refractive index of the sample remains in the same range as those used in the training data, the neural network can accurately predict new features it has never seen such as those demonstrated in Figure 2. If the refractive index of the sample falls outside of the range however, a retraining of the neural network is recommended.

14. In the “Robustness in Low Light” section the authors say: “The continual learning strategy is thus essential to achieve accurate AI inference on new features”

If training must be carried out on the whole dataset to reach maximum quality, it sounds to me as being computationally expensive. What is the training time in this case? How does it compare to iterative phase retrieval? This would convince the readers to consider a switch from iterative to neural phase retrieval.

We thank the reviewer for this comment.

Indeed, the training time scales (almost) linearly with the dataset (see RF_Figure 12), and can become computationally expensive. To tackle this we can choose to limit the maximum number of training data in continual learning to 100000 (i.e. always using the latest 100000 sets of data) which takes roughly 10 min

to train on one ALCF node with 8 A100 GPUs. The optimal number of training data would vary from sample to sample. In our case, training with 10000 dataset (1/10 of the limit) was already sufficient to reach an inference accuracy of 90% as shown in Fig. 5 of the original manuscript. Further increasing the number of training data only marginally improved the accuracy of the neural network.

With 10000 dataset, the training takes about 3 min (see RF_Figure 12), which is still comparability slower than the iterative phase retrieval time for one data set (about 1 min, as stated in the revised manuscript). However, we don't think it is fair to compare these two numbers together. For most part of the experiment, training is performed only at a prolonged interval. Even at the beginning the experiment when training is performed regularly, the phase retrieval speed of our proposed workflow is only limited by the inference speed which, as we have pointed out in the response to comment #1, is about 800 times faster than conventional methods. We have revised the **Discussion** section to clarify this comparison.

15. In the conclusion section the authors say that high-resolution X-ray imaging can be performed using the neural network: "First, by performing AI inference on individual diffraction patterns, the oversampling constraint is eliminated. We show sample image with an accuracy of 90% without any probe overlap between the measured points. This reduces the beam dose by a factor of 6 compared to that required by conventional methods. In the second approach, the beam dose is further reduced by a factor of 10 to 1000 by performing AI inference on low counts data."

I did not see proof of this in the paper. To achieve the desired resolution in ptychography, one must collect sufficient photons at the highest scattering angles incident onto the sensor. To collect the desired photon counts two methods can be used: short exposure/high-overlap scans or long-exposure/lower-overlap scans. In both cases, the total dose deposited onto the sample can be equivalent. You have shown that you can reduce the overlap OR the exposure time, but you cannot claim that you can do both at the same time since the data will have a significantly lower dose. On this note, the low-overlap experiments could have been done with a significantly higher exposure time, making the results better than they should have been. Your results would be more convincing if you would plot the total dose vs resolution/accuracy or if you would be more transparent by stating X-ray dose imparted onto the sample.

We thank the reviewer for pointing this out and gratefully appreciate the nuanced question. In the following we shall address the two methods mentioned by the reviewer: low exposure + high overlap, and long exposure + low overlap.

As shown in the response to comment #3, it is in fact possible to retain the same resolution using low overlap data without increasing the exposure time.

Regarding the 2nd part of the question, it is true that when the exposure time is reduced, the resolution / accuracy would suffer. But the resolution / accuracy cannot be improved even if we increase the overlap. This is again because of the unawareness of the neural network to the concept of overlap.

16. In the X-ray Ptychography Experiment section, I don't see details about the probe size, scanning parameters etc. I think they were mentioned somewhere in the paper, but it would be good to have everything described clearly in a single section.

We thank the reviewer for pointing this out. We have edited the X-ray Ptychography Experiment section to include the missing details. The changes are highlighted in red.

X-ray Ptychography Experiment

X-ray ptychography data was taken at the hard X-ray nanoprobe beamline at the Advanced Photon Source. The photon energy was 10 keV. Two sets of experimental data were acquired. The first set was taken with an

Amsterdam Scientific Instruments Medipix3 detector (516x516 pixels, 55 μm pixel size) sitting at 1.55 m downstream of the sample. The maximum frame rate was 100Hz limited by the data transfer bandwidth of the 1 Gbps network. The second set was taken with a Dectris Eiger2 X 500K detector with 75 μm pixels, located 0.9 m downstream from the sample. Only a subsection of the 128x128 image is used, allowing live inference at the maximum detector frame rate of 2 kHz. A Fresnel Zone Plate with 160 μm diameter and 30 nm outermost zone width was used, and defocused intentionally to reach an about 800 μm spot size. For each scan a piezo motor moves the focusing optics, and hence the beam, in 963 steps following a spiral pattern. The step size varies from 50 nm (for training) to 200 nm while the exposure time varies from 0.4 s (for training) to 0.5 ms.

Reviewer #1 (Remarks to the Author):

I am satisfied with the revisions made in the manuscript. The authors have thoroughly (in my view) addressed the concerns raised in the previous round of review. I recommend acceptance.

Reviewer #3 (Remarks to the Author):

Thank you for the detailed response and changes to the manuscript, which looks clearer to me. A few final comments:

1. I think the following response to reviewer #2 should be featured in the introduction to make it clear that the deep-learning approach is designed more for fast data-streaming rather than to replace existing reconstruction methods in terms of quality. The claims of low-dose, zero overlap imaging seems to suggest that this reconstruction approach would allow replacing existing iterative phase retrieval. However, to achieve truly low-dose imaging you will inevitably need to capture data that cannot be reconstructed with conventional phase retrieval (e.g. zero overlap). However, you also confirm that the approach is designed more for real time visualization rather than high quality data. This to me sounds like a dichotomy, since if conventional iterative phase retrieval is expected to be used for high-resolution data recovery, then a standard dataset will inevitably be captured. Otherwise, as reviewer #2 points out, you will only achieve low-resolution reconstruction at a faster rate.

I am referring to this response:

We thank the reviewer for this comment. It was never our intention to achieve higher accuracy with a ML-based method compared to numerical methods (such as iterative phase retrieval). In fact, it is a common practice to perform a few iterations of numerical optimization after AI-inference, because the former is known for its superior accuracy [1-3]. On the note of accuracy, we also respectfully disagree with the reviewer's claim about ML not recovering fine features in Fig. 2C, Fig. 3E and Fig. 3F. While it is true that the accuracy of the workflow would not exceed that of interactive phase retrieval, we do believe that the ML inference was able to reproduce all the fine details present in those figures. Finally, during a high data rate synchrotron experiment, speed is just as important as accuracy. Scientific users would have all the time they need after the experimental beamtime to perform more accurate (and slower) data analysis, but they need real-time feedback during the experiment to steer the experiment in the right direction. This is particularly true for in situ experiments where samples undergo irreversible changes. The situation is even more challenging with the advent of diffraction limited storage rings (4th generation synchrotrons). The data acquisition speed is dramatically increased with the 100-fold increased brightness, which in turn requires the development of even faster real time data analysis workflows. We believe that this manuscript presents a significant advance by providing a solution to the problem of real-time analysis (and feedback), allowing the use of next generation to their fullest capacity. Please find below the response to the rest of the comments point-by-point.

2. In your revision you stated:

A Fresnel Zone Plate with 160 μm diameter and 30 nm outermost zone width was used, and defocused intentionally to reach an about 800 μm spot size. For each scan a piezo motor moves the focusing optics, and hence the beam, in 963 steps following a spiral pattern. The step size varies from 50 nm (for training) to 200 nm while the exposure time varies from 0.4 s (for training) to 0.5 ms.

I assume the spot size of 800 μm should have said 800nm.

REVIEWERS' COMMENTS

Reviewer #1 (Remarks to the Author):

I am satisfied with the revisions made in the manuscript. The authors have thoroughly (in my view) addressed the concerns raised in the previous round of review. I recommend acceptance.

We thank the reviewer for their critical comments that allowed us to improve the quality of our work.

Reviewer #3 (Remarks to the Author):

Thank you for the detailed response and changes to the manuscript, which looks clearer to me. A few final comments:

1. I think the following response to reviewer #2 should be featured in the introduction to make it clear that the deep-learning approach is designed more for fast data-streaming rather than to replace existing reconstruction methods in terms of quality. The claims of low-dose, zero overlap imaging seems to suggest that this reconstruction approach would allow replacing existing iterative phase retrieval. However, to achieve truly low-dose imaging you will inevitably need to capture data that cannot be reconstructed with conventional phase retrieval (e.g. zero overlap). However, you also confirm that the approach is designed more for real time visualization rather than high quality data. This to me sounds like a dichotomy, since if conventional iterative phase retrieval is expected to be used for high-resolution data recovery, then a standard dataset will inevitably be captured. Otherwise, as reviewer #2 points out, you will only achieve low-resolution reconstruction at a faster rate.

I am referring to this response:

We thank the reviewer for this comment. It was never our intention to achieve higher accuracy with a ML-based method compared to numerical methods (such as iterative phase retrieval). In fact, it is a common practice to perform a few iterations of numerical optimization after AI-inference, because the former is known for its superior accuracy [1-3]. On the note of accuracy, we also respectfully disagree with the reviewer's claim about ML not recovering fine features in Fig. 2C, Fig. 3E and Fig. 3F. While it is true that the accuracy of the workflow would not exceed that of interactive phase retrieval, we do believe that the ML inference was able to reproduce all the fine details present in those figures. Finally, during a high data rate synchrotron experiment, speed is just as important as accuracy. Scientific users would have all the time they need after the experimental beamtime to perform more accurate (and slower) data analysis, but they need real-time feedback during the experiment to steer the experiment in the right direction. This is particularly true for in situ experiments where samples undergo irreversible changes. The situation is even more challenging with the advent of diffraction limited storage rings (4th generation synchrotrons). The data acquisition speed is dramatically increased with the 100-fold increased brightness, which in turn requires the development of even faster real time data analysis workflows. We believe that this manuscript presents a significant advance by providing a solution to the problem of real-time analysis (and feedback), allowing the use of next generation to their fullest capacity. Please find below the response to the rest of the comments point-by-point.

We thank for the reviewer for this comment.

We agree with the reviewer that the distinction made in our response to the 2nd reviewer should be featured in the paper. We have included this in the discussion section. In particular we have discussed the dependence of our approach on iterative phase retrieval as well as the dichotomy pointed out by the reviewer. The revised text is highlighted in red:

We note that the proposed workflow still requires high-overlapped data as well as iterative phase retrieval to produce training data for the neural network, particularly at the beginning of the experiment. As such, a good strategy for imaging beam-sensitive samples is to first run conventional ptychographic measurements in a small area with high overlap and decent count rate. This allows high quality training data to be acquired, which ensures the accuracy of the subsequent low dose imaging on the remaining area with essentially no overlap. The limitation of this strategy is that by removing the overlap between the measured points, we have limited the possibility to perform iterative phase retrieval, and by extension the possibility to perform continual learning.

2. In your revision you stated:

A Fresnel Zone Plate with 160 μm diameter and 30 nm outermost zone width was used, and defocused intentionally to reach and about 800 μm spot size. For each scan a piezo motor moves the focusing optics, and hence the beam, in 963 steps following a spiral pattern. The step size varies from 50 nm (for training) to 200 nm while the exposure time varies from 0.4 s (for training) to 0.5 ms.

I assume the spot size of 800 μm should have said 800nm.

We thank the reviewer for pointing this out.

We have revised the main manuscript to correctly state the size of the defocused probe.